# What does guidance do?
# A fine-grained analysis in a simple setting

**Muthu Chidambaram** *
Duke University
muthu@cs.duke.edu

**Khashayar Gatmiry** *
MIT
gatmiry@mit.edu

**Sitan Chen**†
Harvard University
sitan@seas.harvard.edu

**Holden Lee**†
Johns Hopkins University
hlee283@jhu.edu

**Jianfeng Lu**†
Duke University
jianfeng@math.duke.edu

## Abstract

The use of guidance in diffusion models was originally motivated by the premise that the guidance-modified score is that of the data distribution tilted by a conditional likelihood raised to some power. In this work we clarify this misconception by rigorously proving that guidance fails to sample from the intended tilted distribution. Our main result is to give a fine-grained characterization of the dynamics of guidance in two cases, (1) mixtures of compactly supported distributions and (2) mixtures of Gaussians, which reflect salient properties of guidance that manifest on real-world data. In both cases, we prove that as the guidance parameter increases, the guided model samples more heavily from the boundary of the support of the conditional distribution. We also prove that for any nonzero level of score estimation error, sufficiently large guidance will result in sampling away from the support, theoretically justifying the empirical finding that large guidance results in distorted generations. In addition to verifying these results empirically in synthetic settings, we also show how our theoretical insights can offer useful prescriptions for practical deployment.

## 1 Introduction

With diffusion models having emerged as the leading approach to generative modeling in domains like image, video, and audio [31, 34, 18, 12, 33, 35, 36, 29, 26], there is a pressing need to develop principled methods for modulating their output. For example, how do we impose certain constraints on generated samples, or control their "temperature"? To formalize this, suppose one has access to an unconditional diffusion model approximating the data distribution $p$, as well as a conditional diffusion model approximating the conditional distribution $p(\cdot \mid z)$ for various choices of class labels or prompts $z$.[3] Given $z$ and a parameter $w$, one might wish to sample from the distribution $p$ *tilted* by the conditional likelihood, i.e. the distribution $p^{z,w}$ with density

$$p^{z,w}(x) \propto p(x) \cdot p(z \mid x)^{1+w} .$$

(Throughout, we use lower-case letters to denote densities, and capital letters to denote distributions.)

By varying $w$, we can naturally trade off between diversity and quality: If $w = -1$ then $p^{z,w}$ is the unconditional distribution, if $w = 0$ then $p^{z,w}$ is the conditional distribution $p(\cdot \mid z)$ by Bayes' rule, and as $w \to \infty$, $p^{z,w}$ converges to being supported on the maximizers of the conditional likelihood.

---

*Lead authors, equal contribution

†Equal contribution

[3]In some settings, instead of having access to $p(\cdot \mid z)$, one has access to some model for the conditional likelihood $p(z \mid \cdot)$. The distinction between the "classifier-free" setting [19] and the "classifier" setting is immaterial to this paper, and our theory applies to both.

38th Conference on Neural Information Processing Systems (NeurIPS 2024).

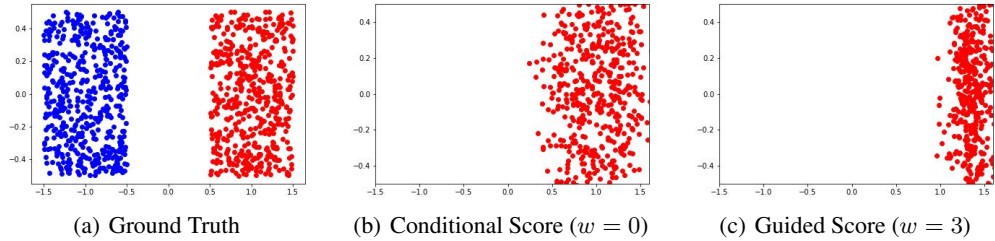

| (a) Ground Truth | (b) Conditional Score ($w = 0$) | (c) Guided Score ($w = 3$) |

Figure 1: We consider sampling from the positive class of a 2D mixture of uniforms (a) using the probability flow ODE with the conditional score (b) and the guided score (c). As can be seen, increasing the guidance weight $w$ clearly biases the distribution of samples to concentrate towards points far away from the other class support.

In practice, the standard approach to try to sample from the tilted distribution is to use *diffusion guidance* [13, 19]. The idea for this is roughly as follows. To sample from a given distribution using diffusion generative modeling, one numerically solves a certain ODE or SDE whose drift depends on the *score function*, i.e. gradient of the log-density, of $p$ convolved with various levels of Gaussian noise. By definition, the score function of $p^{z,w}$ satisfies

$$\nabla \log p^{z,w} = \nabla \log p + (1+w)\nabla \log p(z \mid \cdot) = -w\nabla \log p + (1+w)\nabla \log p(\cdot \mid z). \quad (1)$$

Assuming we have access to approximations of both terms on the right, we can run the corresponding ODE or SDE whose drift can be computed using the above to approximately sample from $p^{z,w}$.

There is however one fundamental snag in the reasoning above. The aforementioned ODE or SDE involves the score function at *different noise levels*, i.e. we would need access to $\nabla \log p_t^{z,w}$, where we use the notation $q_t$ to denote the distribution given by running a certain noising process (see Section 2.1) for time $t$ starting from a distribution $q$. Here $p_t^{z,w}$ means we tilt before adding noise, that is, we take $q = p^{z,w}$ and then apply noise to $q$. Unfortunately, as soon as $t > 0$, the analogue of Eq. (1) no longer holds, i.e.

$$\nabla \log p_t^{z,w} \neq -w\nabla \log p_t + (1+w)\nabla \log p_t(\cdot \mid z). \quad (2)$$

In other words, the operation of applying noise to $p$ and the operation of tilting it in the direction of the conditional likelihood *do not commute*. Nevertheless, in practice it is standard to use the right-hand side of Eq. (2) as an approximation [13, 19]. Sampling using this approximation is called diffusion guidance.

Intriguingly, for appropriate choices of $w$, this heuristic results in generations with high perceptual quality. Yet despite the popularity and empirical success of guidance, our theoretical understanding of this approach is lacking. In this work, we ask:

*What distribution is diffusion guidance actually sampling from?*

**A motivating example.** To see clearly that diffusion guidance is not simply sampling from the tilted distribution, consider the following simple setting. Suppose that there are only two classes $z = -1$ and $z = +1$, and that the corresponding conditional distributions $p(\cdot \mid z = -1)$ and $p(\cdot \mid z = +1)$ have disjoint supports $\Omega_-, \Omega_+$. In this case, the conditional likelihood is simply given by $p(z = i \mid x) \propto \mathbb{1}[x \in \Omega_i]$. In particular, the conditional likelihood is binary-valued, which implies that for any $w > 0$, the tilted distribution $p^{z,w}$ is exactly the same! On the other hand, as Figure 1 shows, increasing $w$ changes the distribution of generated samples to concentrate towards the edge of the support of the guided class.

This simple example already reflects two properties of diffusion guidance that manifest on real-world data as the guidance parameter $w$ increases:

- **Drop in diversity**: The entropy of the distribution over generated samples resulting from diffusion guidance tends towards zero.
- **Divergence towards "archetypes"**: The generated outputs drift more and more to the extreme points in the support of the conditional distribution $p(\cdot \mid z = i)$ to which the diffusion model is

being guided. Note that these extreme points do not necessarily coincide with the set of maximizers of the conditional likelihood, as our example shows.

These phenomena are surprising because we have shown that they can occur even when the conditional likelihood contains *no geometric information* about the data distribution: in our example, all points $x \in \Omega_+$ have the same conditional likelihood under $p(z = +1 \mid x)$, so it is not at all clear why the sampling process should end up preferring points in $\Omega_+$ which are furthest from points in $\Omega_-$.

In this work, we hone in on simple settings in which we can precisely characterize the dynamics of diffusion guidance and explain this counterintuitive behavior. Our main result is to execute such an analysis for mixtures of compactly supported distributions:

**Theorem 1** (Compactly supported setting, informal – see Theorem 4)**.** *Consider a data distribution* $p = \frac{1}{2}p^{(1)} + \frac{1}{2}p^{(-1)}$ *where* $p^{(1)}, p^{(-1)}$ *are* $\beta$-*bounded and supported on disjoint intervals* $[\alpha_1, \alpha_2]$ *and* $[-\alpha_2, -\alpha_1]$ *respectively (see Assumption 1). Suppose that one runs the probability flow ODE with guidance parameter* $w$ *which is larger than some absolute constant. Then with probability* $1 - e^{-\Omega(w)}$, *the resulting sample lies in the interval*

$$\left( \alpha_2 \big( 1 - O(1/\sqrt{\ln w}) \big), \alpha_2 \right), \tag{3}$$

*where the* $O(\cdot)$ *notation hides constants depending on* $\alpha_1, \alpha_2, \beta$.

We also conduct this analysis in a setting where the conditional distributions $p^{(1)}$ and $p^{(-1)}$ do not have compact support by proving an analogous result for mixtures of Gaussians. The proofs in this setting turn out to be somewhat simpler:

**Theorem 2** (Gaussian setting)**.** *Consider the data distribution* $p = \frac{1}{2}\mathcal{N}(1, 1) + \frac{1}{2}\mathcal{N}(-1, 1)$. *Suppose that one runs the probability flow ODE with guidance parameter* $w$ *which is larger than some absolute constant. Then if the resulting sample is denoted by* $\tilde{x}(1)$, *we have*

$$\mathbb{P}(\tilde{x}(1) \geq 0) \geq 1 - e^{-\Omega(w^2)} \qquad \mathbb{P}(\tilde{x}(1) \geq \sqrt{w}) \geq 1 - e^{-\Omega(w)}.$$

Theorems 1 and 2 illustrate that diffusion guidance results in a strong bias towards points in the support of one conditional distribution which are far from points in the support of the other.

These results apply even when the unconditional and conditional diffusion models in question incur zero score estimation error. One shortcoming however is that they fail to corroborate a third commonly observed behavior of guidance in practice:

- **Degradation when guidance is too large**: In practice, even ignoring issues of diversity, there is typically a "sweet spot" for the choice of $w$ such that past that point, the quality of the generated output begins to degrade.

Next, we show how to leverage ideas in the proof of Theorem 1 to explain this degradation. Concretely, we give a simple example where a small perturbation to the score estimate at the tails of the data distribution is enough to take the sampling trajectory given by diffusion guidance far away from the trajectory predicted by Theorem 1:

**Theorem 3.** *Given* $0 < \epsilon < 1$, *assume* $w \geq \widetilde{\Omega}(\sqrt{\log\log(1/\epsilon)})$, *where the hidden constant factor is sufficiently large. There exist densities* $p^{(1)}, p^{(-1)}$ *satisfying the assumptions of Theorem 1, as well as functions* $s_t^{(1)}, s_t$ *satisfying*

$$\|\nabla \log p_t^{(1)} + s_t^{(1)}\|_{L_2(p_t^{(1)})}^2 \leq \epsilon \qquad \text{and} \qquad \|\nabla \log p_t + s_t\|_{L_2(p_t)}^2 \leq \epsilon$$

*such that, if one runs the probability flow ODE with guidance parameter* $w$ *but with* $\nabla \log p_t^{(1)}$ *and* $\nabla \log p_t$ *replaced by* $s_t^{(1)}$ *and* $s_t$ *respectively, then with probability at least* $1 - e^{-\Omega(w)}$, *the resulting sample lies outside of the domain of* $p$.

In other words, for any level of score estimation error, if one takes the guidance parameter $w$ to be too large, the sampler will end up going off the support of the data distribution $p$. Roughly speaking, the idea derives from the proof of Theorem 1. As we will see, one key feature of the guided ODE in the setting of Theorem 1 is that the trajectory first *swings past the edges of the support of* $p$ *and into*

*the tails of the noised data distribution* $p_t$ before returning. As a result, errors in score estimation at these tails can move the sampling process away from the intended trajectory and thus prevent the trajectory from ever returning to the support of $p$, leading to corrupted outputs. We stress that this phenomenon is not an issue of numerical precision: Theorem 3 applies even if one runs diffusion guidance with infinite precision.

Taking inspiration from our theory, we posit that the optimal choice of guidance (from the perspective of sample quality) for compactly supported distributions that approximately satisfy the assumptions of our theory is the largest possible $w$ for which the resulting trajectory does not exhibit this behavior of swinging away from the support of the data distribution and returning. Specifically, we propose a rule of thumb for selecting the guidance strength based on looking at a certain *monotonicity* property of the trajectory and experimentally validate this rule of thumb in both synthetic settings and on image classification datasets. Additionally, for compactly supported distributions that fall outside the scope of Theorem 1, we propose an alternative heuristic based on the ideas of Theorem 3: we should choose the guidance strength as large as possible while still ensuring that final samples are contained within the distribution support. See Section 3 for details.

## 1.1 Related work

It has been observed previously [15, 20] that the score of the tilted distribution convolved with noise is different from what is used in diffusion guidance, i.e. Eq. (2). These works conclude informally that as a result, diffusion guidance should not be sampling from the tilted distribution. In contrast, our work gives rigorous justification for this and provides a fine-grained analysis of the behavior of diffusion guidance on simple toy examples, shedding new light on several key features of the dynamics of guidance.

To our knowledge, only two prior works have sought to theoretically characterize the behavior of guidance, one by Wu et al. [37] and one by Bradley and Nakkiran [3]. Here we discuss the connection to these works in detail and briefly summarize some other relevant results.

**Comparison to Wu et al. [37].** This previous work studied the effect of the guidance parameter $w$ when sampling Gaussian mixture models. They considered two summary statistics: the "classification confidence" and the "diversity" of the generated output.

The former refers to the conditional likelihood $p(z \mid x)$, where $z$ is the index of the component of the Gaussian mixture model to which the sampler is being guided, and $x$ is the generated output. They prove a comparison result showing that the classification confidence of the output of the guided sampler is at least as high as that of the unguided sampler, and they give some quantitative bounds on how much the former exceeds the latter. In particular, they prove that as $w \to \infty$, the classification confidence tends to 1.

As for diversity, they show that the differential entropy of the output distribution of the guided sampler is at most that of the unguided sampler, though they do not provide quantitative bounds on the extent to which the entropy decreases with $w$.

Instead of studying summary statistics of the generated output, we instead give a fine-grained analysis of where exactly the trajectory ends up at different times in the reverse process. While we do not directly study classification confidence, note that in the setting of our main result, Theorem 1, for mixtures of compactly supported product distributions, the statement that classification confidence increases is uninformative because, as mentioned previously, the conditional likelihood for *any point* in the support of the target class is 1. The dynamics that we elucidate in our results can be thought of as a more geometric notion of classification confidence. As for diversity, implicit in our Theorems 1 and 2 are quantitative bounds on how the diversity decreases as $w$ increases.

Additionally, the analysis of how score estimation error impacts diffusion guidance, as well as our empirical findings on real data, are unique to our work.

**Comparison to Bradley and Nakkiran [3].** During the preparation of this manuscript, a very recent theoretical work [3] also studied the extent to which diffusion guidance fails to sample from the tilted distribution. They provided a simple example where the conditional likelihood $p(z \mid x)$ is Gaussian (so that the tilted distribution is also Gaussian) and the probability flow ODE with guidance provably does not sample from the correct tilted distribution. Interestingly, they also study the reverse

*SDE* with guidance and show that it behaves differently under this example than under the probability flow ODE with guidance.

They also observed that diffusion guidance is equivalent to a predictor-corrector scheme where the predictor makes a step according to the conditional distribution $p(x \mid z)$, and the corrector makes a step using Langevin dynamics with respect to the *noised-then-tilted* distribution.

In addition, they considered the mixture of two Gaussians example that we study here. They provide numerical, but non-rigorous evidence that diffusion guidance results in a very different distribution than the tilted one. In contrast, we provide a rigorous analysis of the dynamics proving that this is the case. On the other hand, to our knowledge, the example of a mixture of distributions with compact support has not been considered previously, and the qualitative difference in the behavior of guidance in this setting versus under the mixture of Gaussians setting has not been reported in the literature.

**Sampling guarantees for diffusion models.** Most of the theoretical literature on diffusion models has focused on *unconditional* sampling, e.g., proving that SDE diffusion models can efficiently sample from essentially arbitrary data distributions assuming $L^2$-accurate score estimation [21, 7, 5, 4, 1, 9]. Note the notion of $L^2$ error matches the objective function used in practice. Similar results hold for the ODE under additional smoothness constraints or by using a corrector step [23, 22, 6, 24].

We also mention various recent works on understanding other aspects of diffusion models using mixture models, including provable score estimation [30, 10, 8, 17] and feature emergence [25].

Finally, an unrelated work that touches upon guidance and conditional generation is that of [16]. They give representational bounds on how well conditional score functions can be approximated by ReLU networks in nonparametric settings, which translate to sample complexity bounds for conditional score estimation. In their work, "classifier-free guidance" does not refer to the sampling process that we focus on (indeed, they take guidance parameter $w = 0$ so that the tilted distribution is simply the conditional distribution $p(x \mid z)$). Instead, it refers to the *training* of a neural network that simultaneously parametrizes the unconditional and conditional scores.

## 2 Preliminaries and proof overview

### 2.1 Technical preliminaries

**Mixture models.** We focus on data distributions $p$ which are uniform mixtures of two constituent distributions, taking the form

$$p \triangleq \frac{1}{2}p^{(1)} + \frac{1}{2}p^{(-1)} \, . \tag{4}$$

Throughout, we freely conflate probability measures with their densities. Here $p^{(1)}$ and $p^{(-1)}$ are meant to represent class-conditional distributions, and $p$ is meant to represent the unconditional data distribution.

We will denote a sample from $p$ by the pair $(x, z)$ where $z \in \{\pm 1\}$ specifies the class ($\pm 1$ with probability $\frac{1}{2}$). Given class $z$, the conditional distribution on $x$ is given by $p^{(z)}$.

**Probability flow ODE.** Here we briefly review some basics on diffusion models, specifically the *probability flow ODE*, tailored to the mixture model setting outlined above. Throughout, let $t$ be a time variable which varies from 0 to some terminal time $T$, such that the output of the sampling algorithm is the iterate at time $T$.

To formally introduce the probability flow ODE, we define the parameters $a_t = e^{-T+t}, b_t = \sqrt{1 - a_t^2}$. Let $p_t(\cdot)$ be the distribution of $(a_t x + \xi_t, z)$ where $x \sim p$ is sampled from the mixture and $z$ denotes its class, and $\xi_t \sim N(0, b_t^2)$ is Gaussian noise corresponding to time $t$ of the backward process. Hence, the marginal $p_t(x)$ is the convolution of the target $p$ scaled by $a_t$, and $N(0, b_t^2)$. Denote by $a_*p$ the distribution of $aX$ where $X \sim p$. Then the distribution at time $t$ given $z$ is given by $p_t(\cdot|z = 1) = a_{t*}p^{(1)} \star N(0, b_t^2)$ and $p_t(\cdot|z = -1) = a_{t*}p^{(-1)} \star N(0, b_t^2)$, respectively.

The probability flow ODE with respect to the component $p^{(1)}$ is given by

$$x'(t) = x(t) + \nabla \log p_t(x(t)|z = 1) \, , \tag{5}$$

and analogously for the other component. This ODE has the property that if $x(0)$ is distributed as a sample from $a_{T*}p^{(1)} \star N(0, b_T^2)$, then $x(t)$ is a sample from $a_{(T-t)*}p^{(1)} \star N(0, b_{T-t}^2)$.

**Guidance.** Our goal is to understand the effect of introducing *guidance* into the probability flow ODE (5). Given guidance parameter $w$, the resulting guided ODE is given by

$$x'(t) = x(t) + \nabla \log p_t(x(t)) + (w+1) \log p_t(z|x(t)) \tag{6}$$

$$= x(t) + (w+1)\nabla \log p_t(x(t)|z) - w\nabla \log p_t(x(t)), \tag{7}$$

where in the second step we used Bayes' rule. We will sometimes refer informally to the position of $x(t)$ (or time-reparametrizations thereof) as a *particle*.

Note that when $w = 0$, this is identical to the vanilla probability flow ODE in Eq. (5) for $z = 1$. When $w = -1$, then this is identical to the probability flow ODE for the *unconditional distribution* $p$. Our goal in this work is to understand the behavior of the guided ODE for general $w$, especially large $w$. In particular, as noted at the outset, the "guided score" term $(w+1)\nabla \log p_t(x(t)|z) - w\nabla \log p_t(x(t))$ in Eq. (7) does not correspond to the score function of the tilted distribution convolved with noise, so it is not *a priori* clear what the distribution over the final iterate $x(T)$ actually is.

### 2.2 Intuition for our characterization of the dynamics of guidance

Having formalized the probability flow ODE with guidance, we now provide a high-level overview of our proofs by presenting general intuition for the effect of guidance. The behavior of the guided ODE in the setting of mixtures of compactly supported distributions is the richest, so we focus on illustrating the proof of Theorem 1. In that setting, roughly speaking, we will show that there are three distinct regimes for the evolution of the guided ODE, depending on how the posterior probabilites $p_t(z = -1|x(t))$ and $p_t(z = 1|x(t))$ relate to each other.

First, when the posterior probability $p_t(z = -1|x(t))$ is much larger than $p_t(z = 1|x(t))$, then the score function of the convolved mixture model is dominated by the score of $p^{(-1)}$ convolved with the appropriate Gaussian; in particular, the guided score term in Eq. (7) is almost

$$(w+1)\nabla \log p_t(x(t)|z) - w\nabla \log p_t(x(t)) \approx (2w+1) \log p_t(x(t)|z),$$

i.e. $x(t)$ gets pushed toward the $p^{(1)}$ component with maximum velocity.

The second case is when the posterior probabilities $p_t(z = 1|x(t))$ and $p_t(z = -1|x(t))$ are approximately equal. In this case, the score of the convolved mixture is almost zero since the influences from $p^{(1)}$ and $p^{(-1)}$ cancel each other out. Hence, the guided score term in (7) roughly becomes

$$(w+1)\nabla \log p_t(x(t)|z) - w\nabla \log p_t(x(t)) \approx (w+1)\nabla \log p_t(x(t)|z).$$

We can see that here $x(t)$ will still converge to the right component with high velocity proportional to $w + 1$ in this regime.

Finally the third regime is when $p_t(z = 1|x(t)) > p_t(z = -1|x(t))$, i.e. $x(t)$ is "closer" to $p^{(1)}$. Then the RHS roughly becomes

$$(w+1)\nabla \log p_t(x(t)|z) - w\nabla \log p_t(x(t)) \approx \nabla \log p_t(x(t)|z).$$

In this case $x(t)$ converges to the right component with minimum speed, i.e. proportional to a constant independent of $w$.

Overall, we observe the behavior that when the particle is close to the wrong component, guidance adds more biasing on it to repel it from that component toward the correct one, whereas when the particle is closer to the correct component, it decreases in velocity. This intuitively means that guidance somehow biases the distribution of the correct component to points that are "farther" from the other component, an intuition that we rigorize in Appendices A and B where we prove formal versions of Theorems 1 and 2.

## 3 Experiments

Here we empirically verify the guidance dynamics predicted by Theorems 1 and 2. All experiments in this section were conducted on a single A5000 GPU. We use Jax [2] for the experiments in Section 3.1 and PyTorch [27] for all other experiments.

### 3.1 Synthetic experiments

We first revisit the distribution used in Figure 1 (mixture of uniforms), and then consider the case of mixture of Gaussians in Appendix C.1. The distribution in Figure 1 is constructed by taking $p^{(z)}$ to be $\mathsf{Uniform}([-1/2, 1/2] \otimes [-1/2, 1/2])$ shifted by $2z$ in the $x$-coordinate. Note that although this is a 2-D distribution, the distributions $p^{(1)}$ and $p^{(-1)}$ can be written as $p_1^{(z)} \otimes q$ with $q$ shared, which does not affect the dynamics of Theorem 1. We demonstrated in Figure 1 how sampling with a larger guidance parameter yields a distribution of samples that is more concentrated than the true conditional distribution of the data. We now examine the ODE dynamics that produced these samples and compare them to the dynamics predicted by our theory.

We generate samples using guidance by numerically solving the guided probability flow ODE (7) using the Dormand-Prince method [14] as implemented in JAX [2]. For solving, we use 1000 evaluation steps and take $T = 10$, which we is sufficiently large based on the stipulations of Theorem 1. For obtaining the unconditional and conditional scores necessary for the ODE, it is straightforward to write down exact expressions for this case (which consist of integrals that we can numerically approximate). However, we estimate the scores using a more general approach that can be effectively applied to any mixture distribution for which we can sample both conditionally and unconditionally from.

For brevity, let $A_t$ follow the distribution of $a_t X$, where $a_t$ is defined as before and $X \sim p$ (the mixture distribution). Similarly, let $A_{t,z}$ follow the conditional distribution $a_t X \mid z$. Lastly, letting $p_{b_t \xi}$ denote the density of $\xi_t$, we have the following expressions for the scores:

$$\nabla \log p_t(x) = \frac{\mathbb{E}_{A_t}[\nabla p_{b_t \xi}(x - A_t)]}{\mathbb{E}_{A_t}[p_{b_t \xi}(x - A_t)]}, \tag{8}$$

$$= -\frac{\mathbb{E}_{A_t}\left[\frac{1}{b_t^2} p_{b_t \xi}(x - A_t)(x - A_t)\right]}{\mathbb{E}_{A_t}[p_{b_t \xi}(x - A_t)]}, \tag{9}$$

$$\nabla \log p_t(x \mid z) = \frac{\mathbb{E}_{A_{t,z}}[\nabla p_{b_t \xi}(x - A_{t,z})]}{\mathbb{E}_{A_{t,z}}[p_{b_t \xi}(x - A_{t,z})]}. \tag{10}$$

Both (8) and (10) follow from rewriting the convolutions as expectations and then using dominated convergence to pass the gradient into the expectations. Using the above, we can compute the scores by standard Monte-Carlo.

We use this ODE solving procedure to generate 500 samples from the conditional distribution $p(x \mid z = +1)$ with varying levels of guidance. For each generated sample, we project the computed ODE trajectory on to the $x$-coordinate (as this is the coordinate handled by our theory in this case).

Theorem 1 suggests that as we increase the guidance parameter $w$, the ODE dynamics will push samples farther and farther in the direction of the guided class support before ultimately pulling them back to the support if necessary (i.e. $w$ is large). As we show in Theorem 3, this behavior can be undesirable if the sampling trajectory moves too far away from the desired class support, as it can amplify score estimation errors and lead to issues in the fidelity of the final produced samples.

We can thus intuitively think of increasing the guidance parameter as not only trading off diversity and sample quality, but also trading off stability with sample concentration. This observation indicates that there should be some range of guidance values that allow for the sampling concentration effect while not entering the unstable regime; i.e. those guidance values that do not lead to sampling trajectories that move far away from the guided class support.

To verify this, we plot the mean of the projected ODE trajectories for increasing guidance parameter values alongside the final produced samples from each trajectory in Figure 2. Since large choices of the guidance parameter lead to some trajectories diverging due to numerical instability/score approximation errors, we visualize only the samples and trajectories that were "good" in that they produced final samples constrained within the guided class support (and we indicate this proportion on the plots). The results show that the projected trajectories do indeed follow the predicted dynamics, with larger choices of $w$ leading to a pronounced pullback towards the end of the trajectories.

Furthermore, as suggested earlier, the qualitatively best choices of $w$ appear to correspond to trajectories that do not (significantly) exhibit this pullback effect. In our case, $w = 3$ exhibits the

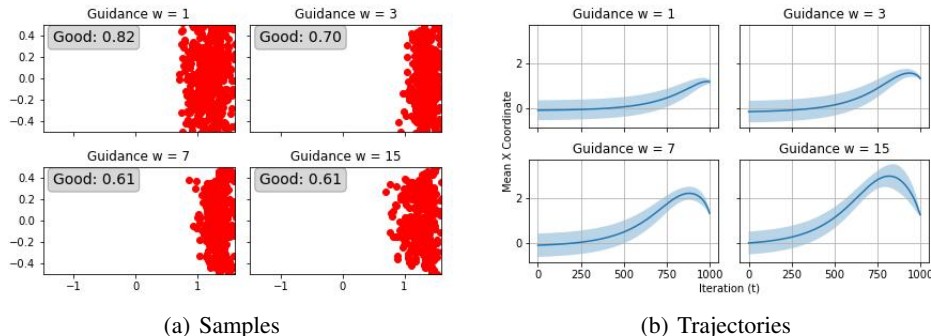

| (a) Samples | (b) Trajectories |

Figure 2: Final samples and mean sampling trajectories produced from solving the probability flow ODE guided towards $y = +1$ in the distribution of Figure 1. The proportion of good samples (i.e. those that were correctly in the class support) is shown with each sample plot, and a 1 standard deviation band is shown around each mean trajectory.

sharpest sample concentration while having significantly better sample fidelity when compared to higher guidance values.

## 3.2 Approximately separable image data

While the synthetic experiments serve to verify our theory, they obviously do not constitute a practical setting in which guidance is used. The most popular use case for guidance in the literature is sampling from image data, and indeed this is what motivated our investigation of guidance for distributions with compact support in the first place.

However, typical image datasets used in the diffusion literature such as ImageNet [11] are known to not be linearly separable, and therefore cannot fall under the exact conditions of Theorem 1. That being said, simpler image datasets are known to be close to linearly separable - in particular, MNIST.

We thus consider using the classifier-free guidance formulation (which corresponds to the second equality in (1)) of [19] to conditionally sample from MNIST with guidance. We use the open-source classifier-free guidance implementation of [28] designed for MNIST.

Although MNIST is perhaps the simplest generative image modeling testbed, it still presents a significant increase in complexity from the synthetic setting. Firstly, compared to the experiments of Section 3.1 and the setting of our theory, we are no longer considering only two classes. Furthermore, there is no guarantee that the class supports are well-separated, or even disjoint. Even more worrying, we do not have access to approximations of the true score functions of the conditional distributions that are guaranteed to be close as in (8) and (10); we have to instead learn a model-based score.

We address the multi-class issue by using the standard one-vs-all reduction. In particular, we fix a single class as the positive class $y = +1$, and then let the union of all other classes represent the negative class $y = -1$. We note that after this reduction, the distribution is close to linearly separable, and we are thus at least close in spirit to maintaining the separation from Theorem 1 under an appropriate basis.

To obtain a projection direction for the sampling dynamics analogous to what was done in Section 3.1, we generate 100 samples from the positive and negative classes using a guidance of $w = 0$, to approximate sampling from the conditional distributions. We then let the projection direction be the difference between the two sample means. For sampling, we use DDPM [18] with 400 time steps and a linear noise schedule, and we found that training the guidance model of [28] for 40 epochs was sufficient to generate high quality samples.

Figure 3 shows the mean projected sampling trajectories alongside the final produced samples for the same choices of guidance parameters used in Figure 2 and the positive class fixed to be the digit 0. We observe the same phenomenon as before: after the guidance parameter $w$ is taken to be sufficiently large, there is a pullback effect in the projected sampling dynamics. Furthermore, once again as before we note that the qualitatively best choice of $w$ (again $w = 3$) is the largest choice for which

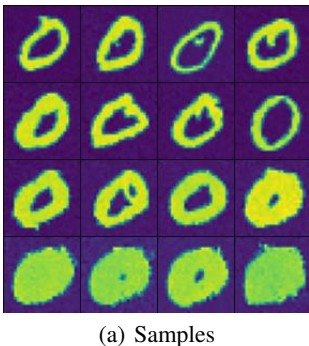
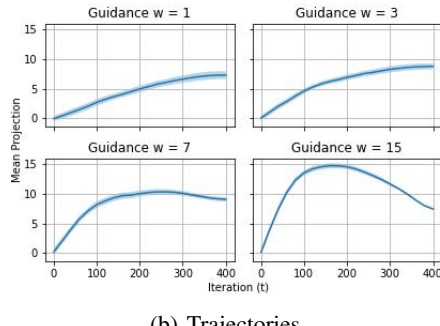

|          (a) Samples          |          (b) Trajectories          |

Figure 3: Final samples and mean projected trajectories produced from sampling using the classifier-free guidance model of [28]. For the positive class, we fix the digit to be 0, and the negative class corresponds to all other digits. Each row of samples from top to bottom corresponds to increasing guidance values.

we can preserve monotonicity of the projected sampling dynamics. These results are not sensitive to the choice of positive class; we show similar plots for every other choice of positive class (i.e. all the non-zero digits) in Appendix C.2. Interestingly, for almost any choice of positive class used in the reduction, the qualitatively optimal choice of guidance amongst the values we consider remains roughly the same.

## 3.3 ImageNet experiments

Although we previously mentioned that experiments on more complicated datasets such as ImageNet are outside the scope of Theorem 1, we show in this section that it is still possible to make qualitative guidance recommendations in such settings based on the ideas of Theorem 3. The idea is that as we scale the guidance parameter $w$ to be large, we start to obtain samples that are no longer within the original data distribution support due to amplification of score/precision errors.

To conduct experiments on ImageNet, we use the *classifier-guided* ImageNet models available from [13]. This is due to the fact that there are no classifier-free guidance models available from [19]. The classifier-guidance formulation corresponds to the first equality in (1). To be consistent with the notation in [13] and to also clearly distinguish the classifier-guided setting from the classifier-free setting of Section 3.2, we will use $s = 1 + w$ throughout the experiments in this section.

First, we illustrate that the behavior exhibited in Figure 3 no longer holds when running diffusion with guidance on ImageNet, at least using the same experimental setup as before. To parallel the experiments of Section 3.2, we use the $256 \times 256$ *conditional* diffusion model released by [13] to generate samples from a fixed ImageNet class (corresponding to $y = +1$ as before), and then use the same model to generate samples from all other classes (corresponding to $y = -1$). We generate 50 samples from the positive and negative classes (due to the cost of sampling at this resolution and the overhead of storing the entire sampling trajectories), and then compute the normalized direction between the two sample means as before.[4] For sampling, we use DDIM [32] with 25 steps, once again because storing the entire sampling trajectories using DDPM with a large number of steps is prohibitive.

For sampling with guidance, we use the *unconditional* diffusion model of [13] with the $256 \times 256$ ImageNet classifier also released by [13]. Note here that [13] combined diffusion guidance with their conditional model for their best results, but this does not fall in to the formulation of (1) and so we use the unconditional model. We use DDIM with 25 steps for the guidance samples as well.

Figure 4 shows the final produced samples alongside the mean projected trajectories for an arbitrarily fixed positive class as in the experiments of Section 3.2. We see that even for extreme guidance scales $s = 25$ the previously observed non-monotonicity phenomenon in the projected trajectories no longer

---

[4]We should note that, in contrast to the MNIST experiments, this choice of direction is very noisy in the ImageNet setting. However, we use the same setup as before both for consistency and to show that Theorem 3 can be applied even in this imperfect experimental setting.

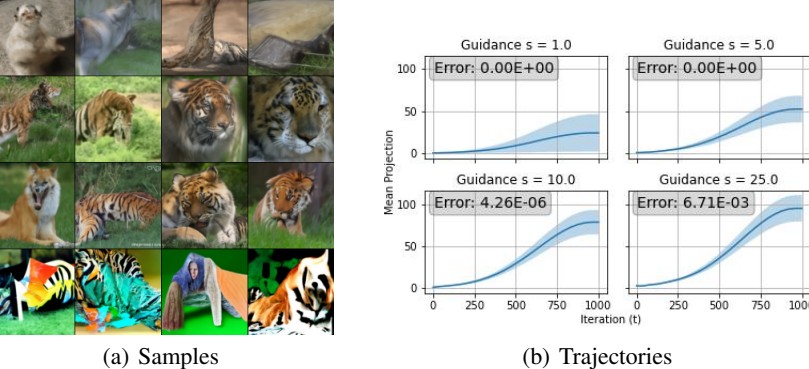

(a) Samples

(b) Trajectories

Figure 4: Final samples and mean projected trajectories produced from sampling using the classifier-guided ImageNet diffusion model of [13]. The positive class here is taken to be 292 (tiger). As before, each row of samples from top to bottom corresponds to increasing guidance values.

occurs. We suspect this can largely be attributed to the fact that the class supports are no longer close to separated, and as a result the direction corresponding to the difference in sample means is no longer a direction for which we can expect the dynamics of Theorem 1 (in fact, we can expect that there is no such direction along which these dynamics occur since the data is not linearly separable even after reducing to two classes). However, we note that as we increase the guidance strength, the final sample correlation along this mean difference direction continues to increase, more akin to the result of Theorem 2.

In tandem with this increasing correlation, we also observe an increase in the mean "support error" of the final samples, which is overlain on to the trajectory plots in Figure 4. This error is computed by taking the mean absolute deviation of every dimension of the final produced samples from the range of valid RGB values $[0, 255]$; dimensions that are outside of this range are truncated so as to form valid images. We find that, at least qualitatively, the largest guidance value ($s = 5$) for which we have no support error seems to perform the best, as taking guidance values larger seems to introduce various visual idiosyncrasies and taking guidance small leads to insufficient concentration on the correct class (as we are guiding an unconditional diffusion model).

We verify that these observations hold for a number of different choices of the positive class; these experiments, along with further discussion of limitations of our experimental setup, are available in Appendix C. We emphasize again that this is merely a minimal demonstration of a possibly useful heuristic, and once again point out that this setting is outside the scope of our theory. Still, an interesting direction for future work could be to run more comprehensive experiments regarding this heuristic (and other heuristics in this section) - such experiments were outside the scope of our available compute resources.

## 4   Conclusion

In this work we gave the first fine-grained analysis of the dynamics of the probability flow ODE with guidance, focusing on two toy settings involving mixture models in one dimension. Our key finding was that not only does the guided ODE fail to sample from the tilted distribution that originally motivated the formulation of guidance, but in fact the guided ODE implicitly leverages geometric information about the data distribution even if such information is absent in the classifier being used for guidance.

Our results open up a number of interesting follow-up directions. For example, our guarantees are restricted to one-dimensional settings, and it would be useful to obtain analogous guarantees for non-trivial high-dimensional settings such as mixtures of bounded densities over disjoint convex bodies. Additionally, we have made no effort to optimize the choice of $w$ in our theoretical guarantees, and it would be interesting to see how small one can take $w$ in theory while still obtain the behavior in our results.

## Acknowledgments

SC thanks David Ding and Yilun Du for illuminating conversations on diffusion guidance. The authors thank Adil Salim for insightful discussions at an earlier stage of this project.

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

# A Mixtures of compactly supported distributions

Consider the case when $p^{(1)}$ and $p^{(-1)}$ are supported on $[\alpha_1, \alpha_2]$ and $[-\alpha_2, -\alpha_1]$, respectively, for parameters $0 < \alpha_1 \leq \alpha_2$. The simplest case is when we are dealing with two points, i.e., when $\alpha_1 = \alpha_2$. However, this case is trivial as adding guidance has no effect: the final distribution with any guidance parameter $w$ remains a point mass on $\alpha_2$.

In contrast, the behavior is unclear if the length of the intervals is nonzero, which is the focus of our results below.

We make the following assumption on $p^{(1)}$ and $p^{(-1)}$:

**Assumption 1.** *We assume the densities* $p^{(1)}, p^{(-1)}$ *have* $\beta$-bounded densities *with respect to each other for some* $\beta \geq 1$. *Namely* $\forall x_1 \in (-\alpha_2, -\alpha_1), x_2 \in (\alpha_1, \alpha_2)$:

$$\frac{1}{\beta} \leq \frac{p^{(-1)}(x_1)}{p^{(1)}(x_2)} \leq \beta.$$

Under Assumption 1 we show that by running the guided ODE starting from Gaussian initialization, we sample from a distribution concentrated at the edge of $p^{(1)}$. In particular, we prove the following result:

**Theorem 4** (Convergence to the edge of the support). *Assume* $\frac{\ln(w)}{16\sqrt{w}} \leq \frac{\alpha_1 \alpha_2}{\beta}$, $\frac{\ln(w)}{\ln \ln(w)} \geq 1024 \left(\frac{\alpha_2}{\alpha_1 \wedge 1}\right)^2$. *Then running the guided ODE with* $T \geq 2 \log 2$ *and parameter* $w$ *for the mixture model* $p$ *defined in* (4) *under Assumption 1, with probability at least* $1 - \exp(-\frac{w \alpha_1^2}{512 \beta^2})$, *the resulting sample lies in the interval*

$$\left(\alpha_2\left(1 - \frac{32}{\sqrt{\ln(w)}}\right), \alpha_2\right). \tag{11}$$

The formal proof of Theorem 4 which combines all the pieces that we present in this section comes at the end of the section. At a high level, we show that the particle goes through two major phases: first, if we have the condition that the initial point satisfies $x(0) \geq -\Theta(-\sqrt{w})$, then it goes towards the support of $p^{(1)}$ and *swings past it*, ending up at position $\Theta(w)$. Then, in the second phase, it comes back to the support of $p^{(1)}$, but by the time it reaches the rightmost edge of the support, it is already close to the end of the process; in particular, we show that it cannot move too much on the support of $p^{(1)}$ once it arrives there. Hence the particle gets stuck near the edge of the support. We formalize this intuition below.

## A.1 Reformulating the ODE

In the setting of this section, the score can be written as

$$\nabla \log p_t(x) = -\frac{\int_{\alpha \in (\alpha_1, \alpha_2) \cup (-\alpha_2, -\alpha_1)} \frac{x - a_t \alpha}{b_t^2} e^{-(x - a_t \alpha)^2 / (2b_t^2)} d\alpha}{\int_{\alpha \in (\alpha_1, \alpha_2) \cup (-\alpha_2, -\alpha_1)} e^{-(x - a_t \alpha)^2 / (2b_t^2)} d\alpha},$$

and

$$\nabla \log p_t(x | z = +1) = -\frac{\int_{\alpha \in (\alpha_1, \alpha_2)} \frac{x - a_t \alpha}{b_t^2} e^{-(x - a_t \alpha)^2 / (2b_t^2)} d\alpha}{\int_{\alpha \in (\alpha_1, \alpha_2)} e^{-(x - a_t \alpha)^2 / (2b_t^2)} d\alpha}.$$

To simplify the notation, for all $t \in [0, T]$, denote the random variable $a_t x + \xi_t$ by $X_t$ (observe that $X_0 = x$). Then by Tweedie's formula, the score at time $t$ can be obtained from the posterior mean of $X_0$ given $X_t$:

$$\nabla \log(p_t(y)) = a_t \mathbb{E}[X_0 - y | X_t = y], \tag{12}$$
$$\nabla \log(p_t(y | z = \pm 1)) = a_t \mathbb{E}[X_0 - y | X_t = y, z = \pm 1]. \tag{13}$$

We denote $\mathbb{E}[X_0 | X_t = y]$ in short by $\mathbb{E}[X_0 | y]$ and $\mathbb{E}[X_0 | X_t = y, z = \pm 1]$ in short by $\mathbb{E}[X_0 | y, z = \pm 1]$. Then, the probability flow ODE can be written as

$$x'(t) = x(t) + \frac{w + 1}{b_t^2} \left(a_t \mathbb{E}[X_0 | x(t), z = 1] - x(t)\right) - \frac{w}{b_t^2} \left(a_t \mathbb{E}[X_0 | x(t)] - x(t)\right).$$

Next, we consider the change of variable $s(t) \triangleq a_t = e^{t-T}$ and its corresponding inverse function $t(s) = \ln(s) + T$, and $\tilde{x}(s) = x(t(s))$, where $s$ varies in the interval $[s_0, 1]$ for $s_0 = e^{-T}$. Define the following conditional probabilities of the two components of the mixture, conditioned on $X_t$:

$$q_t^{(+1)} = \mathbb{P}(z = 1 | X_t = x(t)),$$

$$q_t^{(-1)} = \mathbb{P}(z = -1 | X_t = x(t)).$$

Before proceeding with the proof of Theorem 4, we present a key algebraic manipulation of the probability flow ODE in (7) which enables us to analyze it more easily:

**Lemma 1** (Alternative view on probability flow ODE). *The probability flow ODE can be written as*

$$x'(t) = \frac{1}{b_t^2} \left( a_t \mathbb{E}[X_0 | x(t), z = 1] - a_t^2 x(t) \right) + \frac{w}{b_t^2} q_t^{(-1)} \left( a_t \mathbb{E}[X_0 | x(t), z = 1] - a_t \mathbb{E}[X_0 | x(t), z = -1] \right),$$
(14)

*or with respect to $\tilde{x}(s)$ in variable $s$:*

$$\tilde{x}'(s) = \frac{1}{b_t^2} \left( \mathbb{E}[X_0 | \tilde{x}(s), z = 1] - a_t \tilde{x}(s) \right) + \frac{w}{b_t^2} q_t^{(-1)} \left( \mathbb{E}[X_0 | \tilde{x}(s), z = 1] - \mathbb{E}[X_0 | \tilde{x}(s), z = -1] \right),$$
(15)

*Proof.* Note that combining (7) and (13), we can write

$$x'(t) - x(t) = \frac{w+1}{b_t^2} \left( a_t \mathbb{E}[X_0 | x(t), z = 1] - x(t) \right)$$

$$- \frac{w}{b_t^2} \left( q_t^{(+1)} \left( a_t \mathbb{E}[X_0 | x(t), z = 1] - x(t) \right) + q_t^{(-1)} \left( a_t \mathbb{E}[X_0 | x(t), z = -1] - x(t) \right) \right)$$

$$= \frac{w+1}{b_t^2} \left( a_t \mathbb{E}[X_0 | x(t), z = 1] - x(t) \right) - \frac{w}{b_t^2} \left( a_t \mathbb{E}[X_0 | x(t), z = 1] - x(t) \right)$$

$$+ \frac{w}{b_t^2} q_t^{(-1)} \left( (a_t \mathbb{E}[X_0 | x(t), z = 1] - x(t)) - (a_t \mathbb{E}[X_0 | x(t), z = -1] - x(t)) \right)$$

$$= \frac{1}{b_t^2} \left( a_t \mathbb{E}[X_0 | x(t), z = 1] - x(t) \right) + \frac{w}{b_t^2} q_t^{(-1)} \left( a_t \mathbb{E}[X_0 | x(t), z = 1] - a_t \mathbb{E}[X_0 | x(t), z = -1] \right),$$

which using $a_t^2 + b_t^2 = 1$ implies

$$x'(t) = \frac{1}{b_t^2} \left( a_t \mathbb{E}[X_0 | x(t), z = 1] - a_t^2 x(t) \right) + \frac{w}{b_t^2} q_t^{(-1)} \left( a_t \mathbb{E}[X_0 | x(t), z = 1] - a_t \mathbb{E}[X_0 | x(t), z = -1] \right).$$
(16)

Now changing variable from $t$ to $s(t) \triangleq a_t$, taking the derivative we get $x'(t) = \frac{ds}{dt} \tilde{x}'(s) = a_t \tilde{x}'(s)$. Plugging this into the above, we obtain its equivalent form in variable $s$:

$$\tilde{x}'(s) = \frac{1}{b_t^2} \left( \mathbb{E}[X_0 | \tilde{x}(s), z = 1] - a_t \tilde{x}(s) \right) + \frac{w}{b_t^2} q_t^{(-1)} \left( \mathbb{E}[X_0 | \tilde{x}(s), z = 1] - \mathbb{E}[X_0 | \tilde{x}(s), z = -1] \right). \square$$

### A.2 Analyzing the guided ODE

We begin by sketching our argument in greater detail. First note that the second term in (14), namely

$$\frac{w}{b_t^2} q_t^{(-1)} \left( \mathbb{E}[X_0 | x(t), z = 1] - \mathbb{E}[X_0 | x(t), z = -1] \right),$$
(17)

is a positive dominant term (compared to the first term) unless we have $q_t^{(-1)} = O(1/w)$, which happens only when $x(t) = \Omega(\ln(w))$. First, we show that (1) starting from a high probability region for $x(0)$, the particle will get to $x(t) \geq \Omega(\ln(w))$ at some time $t$, using the dominance of the second term (Lemma 3). Then, in the case where $x(t_0) = \Omega(\ln(w))$ for some time $t_0$, (2) we show a lower bound on the time that it takes for $x(t)$ to get back to the proximity of the origin and then an upper bound on how much it can move inside the support (Lemma 5). Finally in Lemma 6 we show that $x(t)$ does converge to the interval, provided the event of Lemma 3 holds. Building upon these Lemmas, we prove the final result in Theorem 4.

We now proceed with the formal proof. We start with a Lemma showing that as long as $x(t)$ is less than the threshold $\Theta\left(\frac{\log(w)}{\alpha_2}\right)$, then $x'(t)$ is at least of order $\Omega(\sqrt{w})$; hence the particle has a strong push toward the positive direction.

**Lemma 2** (Positive push toward right). *If* $x(t) \leq \frac{\log(w)}{16\alpha_2} \wedge \frac{\alpha_1 \sqrt{w}}{2\beta}$, $\alpha_2^2 \leq \frac{\log(w)}{4}$, *and given that* $\alpha_1 \sqrt{w} \geq \log(w)$, *then for times t s.t.* $a_t^2 \leq \frac{1}{2}$, *we have*

$$x'(t) \geq \frac{a_t \alpha_1 \sqrt{w}}{2\beta b_t^2}.$$

*Proof.* From the assumption $s^2 = a_t^2 \leq \frac{1}{2}$, note that we get $b_t^2 = 1 - s^2 \geq \frac{1}{2}$ and $a_t^2 = s^2 \leq b_t^2$. Hence, for any two points $x_1 \in (-a_t\alpha_2, -a_t\alpha_1)$ and $x_2 \in (a_t\alpha_1, a_t\alpha_2)$, we have

$$
\begin{aligned}
\frac{e^{-(x(t)-x_2)^2/(2b_t^2)}}{e^{-(x(t)-x_1)^2/(2b_t^2)}} &= \frac{e^{-(x(t)-x_1+(x_1-x_2))^2/(2b_t^2)}}{e^{-(x(t)-x_1)^2/(2b_t^2)}} \\
&= e^{(x(t)-x_1)(x_2-x_1)/b_t^2 - (x_2-x_1)^2/(2b_t^2)} \\
&\leq e^{(x(t)+a_t\alpha_2)(x_2-x_1)/b_t^2} \\
&\leq e^{4\alpha_2 x(t) + 2\alpha_2^2 a_t^2/b_t^2} \\
&\leq e^{4\alpha_2(\log(w)/(16\alpha_2)) + 2\alpha_2^2} = \sqrt{w}.
\end{aligned}
$$

Therefore,

$$\frac{\int_{a_t\alpha_1}^{a_t\alpha_2} p^{(1)}(x) e^{-(x(t)-x)^2/(2b_t^2)} dx}{\int_{-a_t\alpha_2}^{-a_t\alpha_1} p^{(-1)}(x) e^{-(x(t)-x)^2/(2b_t^2)} dx} \leq \beta\sqrt{w},$$

which gives

$$\frac{q_t^{(+1)}}{q_t^{(-1)}} = \frac{\mathbb{P}(z = 1 | X_t = x(t))}{\mathbb{P}(z = -1 | X_t = x(t))} \leq \beta\sqrt{w}.$$

Therefore, using $\beta, w \geq 1$,

$$q_t^{(-1)} \geq \frac{1}{1 + \beta\sqrt{w}} \geq \frac{1}{2\beta\sqrt{w}}.$$

Therefore, for the second term in Equation (14) we have

$$\frac{w}{b_t^2} q_t^{(-1)} (a_t \mathbb{E}[X_0 | \tilde{x}(s), z = 1] - a_t \mathbb{E}[X_0 | \tilde{x}(s), z = -1]) \geq \frac{w}{b_t^2} q_t^{(-1)} 2a_t\alpha_1 \geq \frac{a_t\alpha_1\sqrt{w}}{\beta b_t^2}.$$

On the other hand, for the first term we have

$$\frac{1}{b_t^2} \left( a_t \mathbb{E}[X_0 | \tilde{x}(s), z = 1] - a_t^2 x(t) \right) \geq \frac{-a_t^2 x(t)}{b_t^2}.$$

Plugging this into Equation (14) and using the fact that $x(t) \leq \frac{\alpha_1\sqrt{w}}{2\beta}$ and $a_t \leq 1$, we get

$$x'(t) \geq \frac{a_t\alpha_1\sqrt{w}}{2\beta b_t^2}. \qquad \square$$

Next, we show that starting from $\tilde{x}(s_0) \geq -\Theta(\sqrt{w})$, the particle $\tilde{x}(s)$ reaches at least $\Theta(\ln(\omega))$ before time 1. This results from the strong acceleration force toward the right in this phase, coming from the dominance of the aforementioned second term in (14).

**Lemma 3** (First phase). *Assuming* $\frac{\ln(w)}{\alpha_2} \leq \frac{\sqrt{w}\alpha_1}{\beta}$, $T \geq 2\ln 2$, *under the conditions of Lemma 2, given that* $\tilde{x}(s^{(0)}) \geq -\frac{\sqrt{w}\alpha_1}{16\beta}$, *there exists a time* $s_0$ *such that*

$$\tilde{x}(s_0) \geq \frac{\ln(w)}{16\alpha_2},$$

*where recall* $s^{(0)} = e^{-T}$ *is the initial time for the ODE* (15).

*Proof.* Note that as long as $\tilde{x}(s) < \ln(w)/(16\alpha_2)$, Lemma 2 gives

$$\tilde{x}'(s) = \frac{x'(s(t))}{a_t} \geq \frac{\sqrt{w}\alpha_1}{2\beta b_t^2} \geq \frac{\sqrt{w}\alpha_1}{2\beta}$$

which means starting from $\tilde{x}(s^{(0)}) \geq -\frac{\sqrt{w}\alpha_1}{16\beta}$ we definitely reach $\frac{\sqrt{w}\alpha_1}{16\beta}$ by time $s = \frac{1}{2}$ if we continue with the same speed. (Note that the condition $a_t^2 \leq \frac{1}{2}$ of Lemma 2 is satisfied up to time $s = \frac{1}{2}$.) But since $\frac{\ln(w)}{16\alpha_2} \leq \frac{\sqrt{w}\alpha_1}{4\beta}$, then we definitely have to pass the point $\frac{\ln(w)}{16\alpha_2}$. $\qquad\square$

Next, we show that once the particle passes the threshold $\tilde{x}(s) \geq \ln(w)/(4\alpha_2)$, then by time 1 it cannot reach any point much to the left of $\alpha_2$, the edge (right end-point) of the interval. To show this, we need the following helper lemma showing that near the end of the reverse process, the particle is quite close to its conditional denoising.

**Lemma 4** (Conditional Gaussian estimate). *Given $b_t \leq \frac{c_1\alpha_2}{\sqrt{\ln(w)}} \leq 1$ for constant $c_1 \geq 1$ with*

$\frac{\ln(w)}{\ln\ln(w)} \geq 16c_1^2\left(\frac{\alpha_2}{\alpha_1}\right)^2$, *we have*

$$\left|\mathbb{E}[X_0|\tilde{x}(s), z=1] - \tilde{x}(s)\right| \leq \frac{2}{\pi}b_t.$$

*Proof.* We can observe the quantity $\mathbb{E}[X_0|\tilde{x}(s), z = 1] - \tilde{x}(s)$ as the expectation of a gaussian variable, $\tilde{\xi}^{(t)}$, which is conditioned on the union of intervals $(-\alpha_2, -\alpha_1) \cup (\alpha_1, \alpha_2)$, i.e.

$$\mathbb{E}\tilde{\xi}^{(t)} = \frac{\int_{\alpha_1}^{\alpha_2} \frac{1}{\sqrt{2\pi}b_t}e^{(x-\tilde{x}(s))^2/(2b_t^2)}(x - \tilde{x}(s))dx}{\int_{(-\alpha_2,-\alpha_1)\cup(\alpha_1,\alpha_2)} \frac{1}{\sqrt{2\pi}b_t}e^{(x-\tilde{x}(s))^2/(2b_t^2)}} + \frac{\int_{-\alpha_2}^{-\alpha_1} \frac{1}{\sqrt{2\pi}b_t}e^{(x-\tilde{x}(s))^2/(2b_t^2)}(x - \tilde{x}(s))dx}{\int_{(-\alpha_2,-\alpha_1)\cup(\alpha_1,\alpha_2)} \frac{1}{\sqrt{2\pi}b_t}e^{(x-\tilde{x}(s))^2/(2b_t^2)}}.$$

$$(18)$$

For the first integral, we estimate the numerator by half of the integral of absolute value of centered Gaussian with variance $b_t^2$, denoted by $\xi_t$, and the denominator by Gaussian tail bound:

$$\left|\int_{\alpha_1}^{\alpha_2} \frac{1}{\sqrt{2\pi}b_t}e^{(x-\tilde{x}(s))^2/(2b_t^2)}(x - \tilde{x}(s))dx\right| \leq \frac{1}{2}\mathbb{E}|\xi_t| \leq \frac{2}{\pi}b_t. \tag{19}$$

For the denominator,

$$\int_{(-\alpha_2,-\alpha_1)\cup(\alpha_1,\alpha_2)} \frac{1}{\sqrt{2\pi}b_t}e^{(x-\tilde{x}(s))^2/(2b_t^2)}dx \geq \int_{(\alpha_1,\alpha_2)} \frac{1}{\sqrt{2\pi}b_t}e^{(x-\tilde{x}(s))^2/(2b_t^2)}dx \tag{20}$$

$$\geq 1 - 2\int_{\frac{\alpha_1+\alpha_2}{2}}^{\infty} \frac{1}{\sqrt{2\pi}b_t}e^{x^2/(2b_t^2)}dx \tag{21}$$

But using integration by parts:

$$\int_{\frac{\alpha_1+\alpha_2}{2}}^{\infty} \frac{1}{\sqrt{2\pi}b_t}e^{-x^2/(2b_t^2)}dx = \frac{-b_t}{x\sqrt{2\pi}}e^{-x^2/(2b_t^2)}\Big|_{\frac{\alpha_1+\alpha_2}{2}}^{\infty} + \int_{\frac{\alpha_1+\alpha_2}{2}}^{\infty} \frac{b_t}{x^2\sqrt{2\pi}}e^{-x^2/(2b_t^2)}dx$$

$$\leq \frac{b_t}{\frac{\alpha_1+\alpha_2}{2}\sqrt{2\pi}}e^{-\left(\frac{\alpha_1+\alpha_2}{2}\right)^2/(2b_t^2)} + \frac{b_t^2}{\left(\frac{\alpha_1+\alpha_2}{2}\right)^2}\int_{\frac{\alpha_1+\alpha_2}{2}}^{\infty} \frac{1}{\sqrt{2\pi}b_t}e^{-x^2/(2b_t^2)}dx,$$

which implies

$$\int_{-\infty}^{-\alpha_1} \frac{1}{\sqrt{2\pi}b_t}e^{-(x-\tilde{x}(s))^2/(2b_t^2)}dx \leq \frac{1}{1 - 4b_t^2/(\alpha_1+\alpha_2)^2}\frac{b_t}{\frac{\alpha_1+\alpha_2}{2}\sqrt{2\pi}}e^{-\left(\frac{\alpha_1+\alpha_2}{2}\right)^2/(2b_t^2)}$$

$$\leq \frac{1}{1 - 4b_t^2/\alpha_2^2}\frac{2b_t}{\alpha_2\sqrt{2\pi}}e^{-\left(\alpha_2^2/(8b_t^2)\right)}.$$

Therefore, combining with Equation (21),

$$\int_{(-\alpha_2,-\alpha_1)\cup(\alpha_1,\alpha_2)} \frac{1}{\sqrt{2\pi}b_t}e^{(x-\tilde{x}(s))^2/(2b_t^2)}dx \geq 1 - \frac{\frac{2b_t}{\alpha_2}}{1 - \left(\frac{2b_t}{\alpha_2}\right)^2}\frac{1}{\sqrt{2\pi}}e^{-\left(\alpha_2^2/(8b_t^2)\right)}.$$

Now using the assumption on $b_t$,

$$\int_{(-\alpha_2,-\alpha_1)\cup(\alpha_1,\alpha_2)} \frac{1}{\sqrt{2\pi}b_t}e^{(x-\tilde{x}(s))^2/(2b_t^2)}dx \geq 1 - \frac{4c_1}{\sqrt{2\pi\ln(w)}}w^{-\frac{1}{8c_1^2}} \geq \frac{1}{2}, \qquad (22)$$

where the last inequality is due to the fact that $\ln(w) \geq 16c_1^2$. Combining this with Equation (19), we can bound the first term in Equation (18):

$$\left| \frac{\int_{\alpha_1}^{\alpha_2} \frac{1}{\sqrt{2\pi}b_t}e^{(x-\tilde{x}(s))^2/(2b_t^2)}(x-\tilde{x}(s))dx}{\int_{(-\alpha_2,-\alpha_1)\cup(\alpha_1,\alpha_2)} \frac{1}{\sqrt{2\pi}b_t}e^{(x-\tilde{x}(s))^2/(2b_t^2)}} \right| \leq \frac{1}{\pi}b_t. \qquad (23)$$

On the other hand, for the second term in Equation (18), we can upper bound the numerator as

$$\left| \int_{-\alpha_2}^{-\alpha_1} \frac{1}{\sqrt{2\pi}b_t}e^{(x-\tilde{x}(s))^2/(2b_t^2)}(x-\tilde{x}(s))dx \right| \leq 2\alpha_2 \int_{-\alpha_2}^{-\alpha_1} \frac{1}{\sqrt{2\pi}b_t}e^{(x-\tilde{x}(s))^2/(2b_t^2)}dx. \qquad (24)$$

But

$$\int_{(-\alpha_2,-\alpha_1)} \frac{1}{\sqrt{2\pi}b_t}e^{(x-\tilde{x}(s))^2/(2b_t^2)}dx \leq \int_{-\infty}^{-\alpha_1} \frac{1}{\sqrt{2\pi}b_t}e^{(x-\tilde{x}(s))^2/(2b_t^2)}dx$$

$$\leq \int_{2\alpha_1}^{\infty} \frac{1}{\sqrt{2\pi}b_t}e^{-x^2/(2b_t^2)}dx.$$

Using similar integration by part for the tail and the assumption on $b_t$, we get

$$\int_{(-\alpha_2,-\alpha_1)} \frac{1}{\sqrt{2\pi}b_t}e^{(x-\tilde{x}(s))^2/(2b_t^2)}dx \leq \frac{\frac{b_t}{2\alpha_1}}{1-\left(\frac{b_t}{2\alpha_1}\right)^2} \frac{1}{\sqrt{2\pi}}e^{-\left(2\alpha_1^2/(b_t^2)\right)}$$

$$\leq \frac{b_t}{\alpha_1}\frac{1}{\sqrt{2\pi}}w^{-2\alpha_1^2/(c_1^2\alpha_2^2)}$$

$$\leq \frac{b_t}{\sqrt{2\pi}\alpha_1}w^{-2\alpha_1^2/(c_1^2\alpha_2^2)}$$

$$\leq \frac{b_t}{\sqrt{2\pi}\alpha_1 \ln(w)}$$

$$\leq \frac{b_t}{16\sqrt{2\pi}\alpha_2}.$$

Plugging this back into (24)

$$\left| \int_{-\alpha_2}^{-\alpha_1} \frac{1}{\sqrt{2\pi}b_t}e^{(x-\tilde{x}(s))^2/(2b_t^2)}(x-\tilde{x}(s))dx \right| \leq \frac{b_t}{16\sqrt{2\pi}}.$$

Combining this with Equation (22), we obtain the following upper bound for the second term in Equation (18):

$$\left| \frac{\int_{-\alpha_2}^{-\alpha_1} \frac{1}{\sqrt{2\pi}b_t}e^{(x-\tilde{x}(s))^2/(2b_t^2)}(x-\tilde{x}(s))dx}{\int_{(-\alpha_2,-\alpha_1)\cup(\alpha_1,\alpha_2)} \frac{1}{\sqrt{2\pi}b_t}e^{(x-\tilde{x}(s))^2/(2b_t^2)}} \right| \leq \frac{b_t}{8\sqrt{2\pi}}.$$

Finally combining this with our upper bound for the first part in Equation (32) completes the proof. $\qquad \square$

With this helper lemma in hand, we are ready to prove that near the end of the reverse process, the particle does not move much to the left of $\alpha_2$.

**Lemma 5** (Second phase). *For some time $s_0 \in [s^{(0)}, 1]$, suppose $\tilde{x}(s^{(0)}) \geq \frac{\ln(w)}{16\alpha_2}$, $\alpha_2^2 \leq \ln(w)$, and $\frac{\ln(w)}{\ln\ln(w)} \geq 1024\left(\frac{\alpha_2}{\alpha_1 \wedge 1}\right)^2$. Then for all $s \in [s_0, 1]$, we have*

$$\tilde{x}(s) \geq \alpha_2\left(1 - \frac{32}{\sqrt{\ln(w)}}\right).$$

*Proof.* From Equation (15), as long as $\tilde{x}(s) \geq 0$, we have

$$\tilde{x}'(s) \geq \frac{1}{b_t^2} \left( \mathbb{E}[X_0|\tilde{x}(s), z = 1] - a_t \tilde{x}(s) \right) \geq -\frac{a_t x(s)}{b_t^2} = -\frac{a_t \tilde{x}(s)}{1 - s^2} \geq -\frac{\tilde{x}(s)}{1 - s},$$

which implies

$$\ln(\tilde{x}(s)) - \ln(\tilde{x}(s_0)) \geq \ln(1 - s) - \ln(1 - s_0). \tag{25}$$

If we denote by $s_1$ the first time $s \geq s_0$ when $x(s) = a_{t(s)}\alpha_2 = s\alpha_2$, then $s_1$ is certainly larger than the first time that $x(s) = \alpha_2$ as for all $t$, $a_t \leq 1$. Let $s_2$ denote the first time $s \geq s_0$ such that $x(s) = \alpha_2$. From Equation (25),

$$\frac{1 - s_2}{1 - s_0} \leq \frac{\tilde{x}(s_2)}{\tilde{x}(s_0)}$$

which implies

$$s_1 \geq s_2 \geq 1 - \frac{\alpha_2}{\ln(w)/(16\alpha_2)} = 1 - \frac{16\alpha_2^2}{\ln(w)}.$$

But once we reach the interval $(a_t\alpha_1, a_t\alpha_2)$, we can lower bound $x'(s)$ using Lemma 4 In particular, note that $b_{t(s_1)}^2 = 1 - s_1^2 \leq \frac{32\alpha_2^2}{\ln(w)}$. Hence, we can use Lemma 4 with $c_1 = 8$, and given that $\frac{\ln(w)}{\ln \ln(w)} \geq 1024 \left( \frac{\alpha_2}{\alpha_1 \wedge 1} \right)^2$ the conditions of Lemma 4 are satisfied, so we get

$$\frac{1}{b_t^2} \left( \mathbb{E}[X_0|\tilde{x}(s), z = 1] - \tilde{x}(s) \right) \geq \frac{2}{\pi} b_t.$$

Therefore, for $s \geq s_1$,

$$\tilde{x}'(s) \geq \frac{1}{b_t^2} \left( \mathbb{E}[X_0|\tilde{x}(s), z = 1] - a_t \tilde{x}(s) \right)$$

$$\geq \frac{1}{b_t^2} \left( \mathbb{E}[X_0|\tilde{x}(s), z = 1] - \tilde{x}(s) \right)$$

$$= -\frac{2}{\pi} \frac{1}{b_t}$$

$$= -\sqrt{\frac{4}{\pi^2(1 - s^2)}}$$

$$\geq -\sqrt{\frac{4}{\pi^2(1 - s)}}.$$

On the other hand, note that from our definition of $s_1$:

$$\tilde{x}(s_1) = \alpha_2 s_1 \geq \alpha_2 \left( 1 - \frac{16\alpha_2^2}{\ln(w)} \right).$$

Therefore, using $\alpha_2^2 \leq \ln(w)$,

$$\tilde{x}(1) \geq \tilde{x}(s_1) - \int_{s_1}^1 \sqrt{\frac{4}{\pi^2(1 - s)}}$$

$$= \tilde{x}(s_1) - \sqrt{\frac{16(1 - s_1)}{\pi^2}}$$

$$\geq \alpha_2 \left( 1 - \frac{16\alpha_2^2}{\ln(w)} \right) - 16\sqrt{\frac{\alpha_2^2}{\pi^2 \ln(w)}}$$

$$\geq \alpha_2 - \frac{32\alpha_2}{\sqrt{\ln(w)}}. \qquad \square$$

Next, we show that when the process gets close to the end, if the particle is on the right side of both of the intervals, then the first term in Equation (14) will dominate the second term and the particle is most likely to converge to some point in the interval $(\alpha_1, \alpha_2)$.

**Lemma 6** (Convergence to the support). *Assume $\alpha_2 - \alpha_1 \geq \frac{16}{\sqrt{\ln w}}$. For $s_0 \geq \frac{3}{4}$ suppose $\tilde{x}(s) \geq \frac{\alpha_1 + \alpha_2}{2}$ for all $s \in [s_0, 1]$. Then, for any $s \in [s_1, 1]$ where*

$$s_1 = \left( 1 - \frac{16(1-s_0)^5}{(1-2(1-s_0))^4 \, (x(s_0) - \alpha_2)^4} \right) \vee s_0,$$

*we have*

$$\tilde{x}(s) \leq \alpha_2 \left( 1 + 4(1-s_0) \left( 1 + \frac{\alpha_2 w}{\alpha_1^2} \right) \right).$$

*Proof.* First note that for every $x_1 \in (-a_t\alpha_2, -a_t\alpha_1)$ and $x_2 \in (a_t\alpha_1, a_t\alpha_2)$ when $s \geq \frac{1}{2}$,

$$\frac{e^{-(x(t)-x_2)^2/(2b_t^2)}}{e^{-(x(t)-x_1)^2/(2b_t^2)}} = \frac{e^{-(x(t)-x_1+(x_1-x_2))^2/(2b_t^2)}}{e^{-(x(t)-x_1)^2/(2b_t^2)}}$$

$$= e^{(x(t)-x_1)(x_2-x_1)/b_t^2 - (x_2-x_1)^2/(2b_t^2)}.$$

But since $x(t) \geq \frac{\alpha_1 + \alpha_2}{2}$, we have

$$\frac{(x(t)-x_1)(x_2-x_1)}{b_t^2} \geq \frac{(a_t\alpha_1 + a_t\alpha_2 - 2x_1)(x_2-x_1)}{2b_t^2}$$

$$\geq \frac{(x_2-x_1)^2}{2b_t^2} + \frac{(a_t\alpha_1 - x_1)(x_2-x_1)}{2b_t^2}$$

$$\geq \frac{4a_t^2\alpha_1^2}{2b_t^2}.$$

Therefore

$$\frac{e^{-(x(t)-x_2)^2/(2b_t^2)}}{e^{-(x(t)-x_1)^2/(2b_t^2)}} \geq e^{4s^2\alpha_1^2/(2(1-s^2))}$$

$$= e^{4s^2\alpha_1^2/(2(1-s)(1+s))}$$

$$\geq e^{\alpha_1^2/4(1-s)}.$$

Therefore

$$q_t^{(-1)} \leq e^{-\alpha_1^2/4(1-s)},$$

which implies

$$\left| \frac{w}{b_t^2} q_t^{(-1)} \left( a_t \mathbb{E}[X_0|x(t), z=1] - a_t\mathbb{E}[X_0|x(t), z=-1] \right) \right| \leq \frac{we^{-\alpha_1^2/4(1-s)}}{1-s^2} (2s\alpha_2)$$

$$\leq \alpha_2 w \frac{e^{-\alpha_1^2/4(1-s)}}{1-s}.$$

But integrating this upper bound from $s_0$ to 1 by changing variable $\kappa = \frac{1}{1-s}$ (hence $d\kappa = -\frac{1}{(1-s)^2} ds$)

$$\int_{s_0}^1 \alpha_2 w \frac{e^{-\alpha_1^2/4(1-s)}}{1-s} ds = \int_{\kappa_0}^1 \alpha_2 w \frac{1}{\kappa} e^{-\alpha_1^2 \kappa/4} d\kappa$$

$$\leq \frac{\alpha_2 w}{\kappa_0} \int_{\kappa_0}^1 e^{-\alpha_1^2 \kappa/4} d\kappa$$

$$= \frac{4\alpha_2 w}{\alpha_1^2 \kappa_0}$$

$$= \frac{4\alpha_2 w}{\alpha_1^2} (1 - s_0). \tag{26}$$

On the other hand, for the first term in Equation (15), as long as $\tilde{x}(s) \geq \frac{1}{2s_0-1}\alpha_2$, we get

$$\alpha_2 - s_0\tilde{x}(s) \leq \frac{\alpha_2 - \tilde{x}(s)}{2}.$$

Therefore,

$$\frac{1}{b_t^2}\left(\mathbb{E}[X_0|\tilde{x}(s), z=1] - a_t\tilde{x}(s)\right) \leq \frac{1}{1-s^2}(\alpha_2 - s_0\tilde{x}(s))$$

$$\leq \frac{1}{2(1-s^2)}(\alpha_2 - \tilde{x}(s))$$

$$\leq \frac{1}{4(1-s)}(\alpha_2 - \tilde{x}(s)). \tag{27}$$

Now let

$$\phi_1(s,x) = \frac{1}{b_t^2}\left(\mathbb{E}[X_0|x+\alpha_2, z=1] - a_t(x+\alpha_2)\right), \tag{28}$$

$$\phi_2(s,y) = -\frac{1}{4(1-s)}y, \tag{29}$$

and define the following ODEs for $s \in [s_0, 1]$:

$$B'(s) = \phi_1(s, B(s)),$$
$$d'(s) = \phi_2(s, d(s)),$$

with

$$B(s_0) = d(s_0) = \tilde{x}(s_0) - \alpha_2.$$

Note that using (27) and (29), we have

$$\phi_1(s,x) \leq \phi_2(s,x).$$

Define the integral of the first term as

$$A(s_1) \triangleq \int_{s_0}^{s_1} \frac{1}{b_t^2}\left(\mathbb{E}[X_0|\tilde{x}(s), z=1] - a_t\tilde{x}(s)\right)ds = \int_{s_0}^{s_1} \phi_1(s, \tilde{x}(s))ds, \tag{30}$$

First comparing Equations (15) and (30), since the second term in the RHS of the ODE in (15) is always positive, from ODE comparison theorem we get $B(s) \geq \tilde{x}(s_0) + A(s)$.

Therefore, assuming $\tilde{x}(s) \geq \frac{1}{2s_0-1}\alpha_2$ and letting $s_1' \geq s_0$ be the first time that $B(s_1') = \frac{1}{2s_0-1}\alpha_2$, by ODE comparison theorem we have $B(s_1') - B(s_0) \leq d(s_1') - d(s_0)$. Hence

$$A(s_1') \leq B(s_1') - \tilde{x}(s_0) = B(s_1') - B(s_0) \leq d(s_1') - d(s_0). \tag{31}$$

But we can solve the ODE in (29):

$$\ln\left(\frac{d(s_1)}{d(s_0)}\right) = \frac{1}{4}\ln\left(\frac{1-s_1}{1-s_0}\right).$$

Note that inequality (31) is valid more generally up to time $s_1'$, when $B(s)$ reaches $\frac{1}{2s_0-1}\alpha_2 - \alpha_2 = \frac{2(1-s_0)}{1-2(1-s_0)}\alpha_2$. Now we solve for the value of $\tilde{s}_1$ when $d(\tilde{s}_1)$ reaches $\frac{2(1-s_0)}{1-2(1-s_0)}\alpha_2$, which upper bounds $s_1'$ due to (31):

$$\frac{\frac{2(1-s_0)}{1-2(1-s_0)}\alpha_2}{x(s_0) - \alpha_2} = \left(\frac{1-\tilde{s}_1}{1-s_0}\right)^{1/4}.$$

Therefore

$$s_1' \leq \tilde{s}_1 = 1 - \frac{16(1-s_0)^5}{(1-2(1-s_0))^4 (x(s_0)-\alpha_2)^4},$$

and from definition for this choice of $\tilde{s}_1$ we get from Equation (31)

$$A(\tilde{s}_1) \leq d(s_1) - d(s_0) = \frac{1}{2s_0-1}\alpha_2 - \alpha_2 - (\tilde{x}(s_0) - \alpha_2) = \frac{2(1-s_0)}{2s_0-1}\alpha_2 - (\tilde{x}(s_0) - \alpha_2). \tag{32}$$

On the other hand, note that for any time $s_2 \geq s_1'$ before $\tilde{x}(s)$ reaches $\frac{\alpha_1 + \alpha_2}{2}$ (if it ever reaches that value), the first term in (15) is always non-positive. Hence, we have

$$A(s_2) \leq \frac{2(1 - s_0)}{2s_0 - 1} \alpha_2 - (\tilde{x}(s_0) - \alpha_2). \tag{33}$$

Note that the inequality (33) is true even when $\tilde{x}(s_0) \leq \frac{1}{2s_0 - 1} \alpha_2$ as the right hand side is positive and the left hand side is negative in (33) in this case. Hence, overall we showed that for any time $s_2 \geq \tilde{s}_1 \vee s_0$, as long as $\tilde{x}$ has not reached $\frac{\alpha_1 + \alpha_2}{2}$, we have (33).

Finally combining the upper bounds on the first and second terms in RHS of the ODE in (15) that we derived in Equations (26) and (32):

$$\tilde{x}(s_2) \leq \tilde{x}(s_0) + A(s_2) + \frac{4\alpha_2 w}{\alpha_1^2} (1 - s_0)$$

$$\leq \tilde{x}(s_0) + \frac{2(1 - s_0)}{2s_0 - 1} \alpha_2 - (\tilde{x}(s_0) - \alpha_2) + \frac{4\alpha_2 w}{\alpha_1^2} (1 - s_0)$$

$$\leq \alpha_2 \left( 1 + (1 - s_0) \left( \frac{2}{1 - 2(1 - s_0)} + \frac{4\alpha_2 w}{\alpha_1^2} \right) \right)$$

$$\leq \alpha_2 \left( 1 + 4(1 - s_0) \left( 1 + \frac{\alpha_2 w}{\alpha_1^2} \right) \right).$$

where we used $s_0 \geq \frac{3}{4}$. This completes the proof. $\qquad \square$

Next, combining all the pieces, we prove Theorem 4.

*Proof of Theorem 4.* First, note that we sample the initial point $x(s_0)$ according to $\mathcal{N}(0, 1)$, hence with probability at least $1 - e^{-\frac{w\alpha_1^2}{512\beta^2}}$ we have

$$\tilde{x}(s_0) \geq -\frac{\sqrt{w}\alpha_1}{16\beta}.$$

Then, from Lemma 3, we get that for some time $s_0 \leq 1$,

$$\tilde{x}(s_0) \geq \frac{\ln(w)}{16\alpha_2}.$$

Let $s_0$ be the minimum such time. Now plugging this into Lemma 5 then implies for all $s \geq s_0$

$$\tilde{x}(s) \geq \alpha_2 \left( 1 - \frac{32}{\sqrt{\ln(w)}} \right). \tag{34}$$

Now take $\tilde{s}_0 \geq s_0 \vee \frac{3}{4}$. Then we can use Lemma 6 with $s_0 = \tilde{s}_0$ because $\alpha_2 \left( 1 - \frac{32}{\sqrt{\ln(w)}} \right) \geq \frac{\alpha_1 + \alpha_2}{2}$ so its condition is satisfied from (34); then, Lemma 6 implies that there exists a time $s_1$ such that for all $s \in [s_1, 1]$:

$$\tilde{x}(s) \leq \alpha_2 \left( 1 + 4(1 - \tilde{s}_0) \left( 1 + \frac{\alpha_2 w}{\alpha_1^2} \right) \right). \tag{35}$$

Note that $\tilde{s}_0$ can be picked any value in the interval $(s_0 \vee \frac{3}{4}, 1)$. Therefore, picking $\tilde{s}_0 \to 1$, Equations (34) and (35) show that with probability at least $1 - e^{-\frac{w\alpha_1^2}{8\beta^2}}$, $\tilde{x}(s)$ converges to the interval $\left( \alpha_2 \left( 1 - \frac{32}{\sqrt{\ln(w)}} \right), \alpha_2 \right)$. $\qquad \square$

## B  Mixtures of Gaussians

In this section we adapt our analysis to the setting of mixtures of two equal-variance Gaussians. Given mean parameter $\mu$ and variance $\sigma_0^2$, consider the random variables

$$z \sim \mathsf{Uniform}(\{\pm 1\})$$
$$x \sim \mathcal{N}(z\mu, \sigma_0^2).$$

The random variable $x$ is distributed according to a mixture of two Gaussians. Let $p_0$ denote its density.

Whereas in the case of mixtures of compactly supported distributions, we found that the guided ODE converges to the edge of the support of one of the components, in the case of mixtures of Gaussians we find that as the guidance parameter increases, the resulting sample moves further and further towards infinity at a rate that scales with $\Theta(\sqrt{w})$. This is formalized in the following main result of this section. For convenience, we take $\mu = 1$ and $\sigma_0 = 1$, but our analysis naturally extends to other choices for these parameters.

**Theorem 5.** *For $w$ larger than some absolute constant (see Lemmas 7 and 8), running the guided ODE with parameter $w$ for the mixture $\frac{1}{2}\mathcal{N}(1,1) + \frac{1}{2}\mathcal{N}(-1,1)$ results in a sample $\tilde{x}(1)$ for which*

$$\mathbb{P}(\tilde{x}(1) \geq 0) \geq 1 - e^{-\Omega(w^2)}$$
$$\mathbb{P}(\tilde{x}(1) \geq \sqrt{w+1}/4) \geq 1 - e^{-\Omega(w)}.$$

The first bound implies that an overwhelming fraction of the probability mass in the output distribution is given by positive values. In fact, our proof says more. Based on where the particle is initialized, there is a bifurcation in where the guided ODE sends the particle: points to the left of $-2w - 1$ do not result in positive samples, whereas points to the right of $-2w - \Theta(\ln w)$ result in positive samples.

The second bound in Theorem 5 gives a qualitatively stronger guarantee at the cost of a weaker high-probability bound: not only is the overwhelming majority of the output distribution concentrated on positive values, but most of that mass is concentrated on values at least $\Omega(\sqrt{w})$.

### B.1 Reformulating the ODE

We begin by performing some preliminary calculations to simplify the guided ODE, culminating in the simplified expression in Equation (39) below.

Recall that $p_t(\cdot)$ denotes the distribution of $(a_t x + \xi_t, z)$ where $x \sim p$ is sampled from the mixture and $z$ denotes its class, and $\xi_t \sim N(0, b_t^2)$ for $a_t = e^{-T+t}$ and $b_t = \sqrt{1 - a_t^2}$. We will often omit the subscript $t$ when referencing $a_t$ and $b_t$ when the context is clear. Hence for $\xi \sim N(0, I)$,

$$p_t(y) \propto \exp\left(\frac{-(y - a\mu)^2}{2(a^2\sigma_0^2 + b^2)}\right) + \exp\left(\frac{-(y + a\mu)^2}{2(a^2\sigma_0^2 + b^2)}\right).$$

Let $\sigma_t = a^2\sigma_0^2 + b^2$. Then a straightforward calculation shows that

$$\nabla \log p_t(y) = \frac{1}{\sigma_t^2}\left(-y + a\mu \tanh\left(\frac{a\mu y}{\sigma_t^2}\right)\right).$$

We also have

$$p_t(z|y) = \frac{\exp\left(\frac{-(y - za\mu)^2}{2(a^2\sigma_0^2 + b)}\right)}{\exp\left(\frac{-(y - a\mu)^2}{2\sigma_t^2}\right) + \exp\left(\frac{-(y + a\mu)^2}{2\sigma_t^2}\right)} = \frac{\exp\left(\frac{za\mu y}{\sigma_t^2}\right)}{\exp\left(\frac{a\mu y}{\sigma_t^2}\right) + \exp\left(-\frac{a\mu y}{\sigma_t^2}\right)}$$

$$\nabla \log p_t(z|y) = \frac{a\mu}{\sigma_t^2}\left(z - \tanh\left(\frac{a\mu y}{\sigma_t^2}\right)\right).$$

Then

$$\nabla \log p_t(y) + (w+1)\nabla \log p_t(z|y) = \frac{1}{\sigma_t^2}\left((w+1)za\mu - y - wa\mu \tanh\left(\frac{a\mu y}{\sigma_t^2}\right)\right).$$

As a sanity check, for $w = 0$, this gives $\frac{1}{\sigma_t^2}(za\mu - y)$, which is the score for $N(za\mu, \sigma_t^2)$.

The probability flow ODE for the guided model is then

$$x'(t) = x(t) + \frac{1}{\sigma_t^2}\left((w+1)za_t\mu - x(t) - wa_t\mu \tanh\left(\frac{a_t\mu x(t)}{\sigma_t^2}\right)\right)$$

$$= x(t) + \frac{1}{\sigma_t^2}\left((za_t\mu - x(t)) + wa_t\mu\left(z - \tanh\left(\frac{a_t\mu x(t)}{\sigma_t^2}\right)\right)\right), \qquad (36)$$

where $x(0) \sim N(0, I)$ and $t$ varies in $[0, T]$. For simplicity we assume $\mu = \sigma_0 = 1$. Then, writing the guidance equation for this mixture in the form (7):

$$x'(t) = x(t) - (w + 1)(x(t) - a_t) + w(x(t) - a_t \tanh(a_t x(t))).$$

Now changing variables to $s = e^{-T+t}$, $\bar{x}(s) = x(\ln(s) + T)$, we can rewrite the ODE in terms of $\bar{x}(s)$: (Note that we have $s\bar{x}'(s) = x'(t)$)

$$s\bar{x}'(s) = \bar{x}(s) - (w + 1)(\bar{x}(s) - s) + w\Big(\bar{x}(s) - s \tanh(s\bar{x}(s))\Big),$$
$$= s(w + 1) - sw\Big(\tanh(s\bar{x}(s))\Big),$$

which boils down to

$$\bar{x}'(s) = (w + 1) - w \tanh(s\bar{x}(s)), \tag{37}$$

for $s \in [s_0, 1]$, $s_0 = e^{-T}$, with initial condition $\bar{x}(s_0) = x(0) = x$.

Now sending $T \to \infty$, the interval $[s_0, 1]$ converges to $[0, 1]$, and we define the corresponding ODE in variable $\tilde{x}(s)$:

$$\tilde{x}(0) = x, \tag{38}$$
$$\tilde{x}'(s) = (w + 1) - w \tanh(s\tilde{x}(s)), \tag{39}$$

where $x \sim \mathcal{N}(0, 1)$. Below, we study the behavior of the ODE in (39) for different initial conditions $\tilde{x}(0) = x$.

## B.2 Analyzing the guided ODE

The proof of Theorem 5 is based on two key results which break down the behavior of the guided ODE dynamics into two cases depending on where the initialization lies.

First, Lemma 7 below controls how far $\tilde{x}(1)$ moves to the right when the initialization $\tilde{x}(0) = x$ is in the interval $[-2w + \Theta(\ln w), 0]$. This gives rise to the first bound in Theorem 5. Lemma 8 below handles the case when the initialization is in the interval $[-\Theta(\sqrt{w}), 0]$, giving rise to the second bound in Theorem 5. In the first case, we show that the movement of $\tilde{x}(t)$ can be as large as $\Theta(w)$ while in the second lemma we show movement of order $\Theta(\sqrt{w})$.

**Lemma 7** (Characterizing when the particle moves to the positive side)**.** *Suppose $w \geq 100$. Then provided that $\tilde{x}(0) > -2w + 26 \ln(w)$, we have $\tilde{x}(1) \geq 0$.*

Before proceeding with the proof, we note that up to the $\Theta(\ln w)$ term, this analysis is nearly tight in the regime of large $w$. The reason is that if $\tilde{x}(0) < -2w - 1$, then because the velocity in Equation (39) is upper bounded by $2w + 1$ at all times, the particle will remain negative at all times.

*Proof.* First, note that the right-hand side of Equation (39) is lower bounded by 1, so if $\tilde{x}(0) \geq 0$, then $\tilde{x}(1) \geq \tilde{x}(0) + w \geq 0$. More generally, this implies that as soon as the particle becomes positive, it continues to move to the right at rate lower bounded by 1.

Next, we handle the case of $\tilde{x}(0) < 0$. Observe that $w \geq 100$ implies $w \geq 18 \ln(w) + 2$. Moreover, for $\tilde{x}(s) \leq 0$ we have $\tilde{x}'(s) \geq w + 1$. If $\tilde{x}(0) \geq -w$, then we are done as this implies that the particle moves at rate at least $w + 1$ to the right until it becomes positive, at which point it remains positive by the argument in the previous paragraph.

In the rest of the proof, we assume $x < -w$. Define $c = \ln(w)$. Now for the first $s \in [0, c/(w + 1)]$ window of time, using the fact that $-1 \leq \tanh(s\tilde{x}(s)) \leq 0$, we get

$$\tilde{x}(c/(w + 1)) \geq x + (w + 1)\frac{c}{w + 1} = x + c$$
$$\tilde{x}(c/(w + 1)) \leq x + (2w + 1)\frac{c}{w + 1} \leq x + 2c,$$

where the last inequality is due to the definition of $x$ and the fact that $w \geq 18c + 2 \geq 6c + 1$. From this, we see that for time $s_0 = c/(w + 1)$,

$$s_0 \tilde{x}(s_0) \leq -c.$$

Now let $s_1 \geq s_0$ be the first time (if any) where

$$s_1 \tilde{x}(s_1) = -c.$$

In the following we upper bound $s_1$. Defining $\epsilon$ by $\epsilon/w = 1 + \tanh(-c)$, we see from the definition of ODE (39) that for all $s_0 \leq s \leq s_1$:

$$\tilde{x}'(s) \geq 2w + 1 - \epsilon.$$

Therefore, we get for all times $s_0 \leq s \leq s_1$:

$$
\begin{aligned}
0 \geq \tilde{x}(s) &\geq \tilde{x}(c/(w+1)) + (2w + 1 - \epsilon)\left(s - \frac{c}{w+1}\right) \\
&\geq x + c + (2w + 1 - \epsilon)\left(s - \frac{c}{w+1}\right) \geq x - c + (2w + 1 - \epsilon)s.
\end{aligned}
\tag{40}
$$

This means from the definition of $s_1$:

$$(x - c + (2w + 1 - \epsilon)s_1)s_1 \leq -c.$$

Therefore, if $s^*$ is the larger zero of the following quadratic function (in variable $s$),

$$(x - c + (2w + 1 - \epsilon)s)s + c,$$

then $s_1 \leq s^*$. Now we estimate $s^*$ by completing the square:

$$s^2 - \frac{-x + c}{2w + 1 - \epsilon}s + \frac{c}{2w + 1 - \epsilon} = 0,$$

which implies

$$\left(s^* - \frac{-x + c}{2(2w + 1 - \epsilon)}\right)^2 = \left(\frac{-x + c}{2(2w + 1 - \epsilon)}\right)^2 - \frac{c}{2w + 1 - \epsilon} \leq \left(\frac{-x + c}{2(2w + 1 - \epsilon)} - \frac{c}{-x + c}\right)^2.
\tag{41}$$

This means

$$s_1 \leq s^* \leq \frac{-x + c}{2w + 1 - \epsilon} - \frac{c}{-x + c}.
\tag{42}$$

Using this bound on $s_1$ we want to show that $\tilde{x}(1) \geq 0$. Indeed, it is enough to show that for time $s^*$ which is an upper bound on $s_1$, for the remainder of the time $(1 - s_1)$, the movement of $\tilde{x}(s^*)$, which is at least $(1 - s^*)(w + 1)$, is at least $-\tilde{x}(s^*)$. First note that

$$(1 - s^*)(w + 1) \geq \left(\frac{2w + 1 - \epsilon + x - c}{2w + 1 - \epsilon} + \frac{c}{-x + c}\right)(w + 1),$$

and using the inequalities $2w + 1 - \epsilon \leq 2w + 2$ and $-x + c \leq 2w$ we get

$$(1 - s^*)(w + 1) \geq \frac{2w + 1 - \epsilon + x - c}{2} + \frac{c}{2}.
\tag{43}$$

On the other hand, note that from (37), $\tilde{x}'(s) \leq (2w + 1)$, we have $\tilde{x}(s) \leq \tilde{x}(s_0) + (2w + 1)(s - s_0)$. In particular, from the definition of $s_1$,

$$s_1\left(\tilde{x}(s_0) + (2w + 1)(s_1 - s_0)\right) \geq -c.
\tag{44}$$

In particular, consider the following quadratic

$$Q(s) = (2w + 1)s^2 + s\left(\tilde{x}(s_0) - (2w + 1)s_0\right) + c = 0,
\tag{45}$$

and let its roots be $s_0^* < s_1^*$. First note that $Q(0) = c > 0$ and $Q(s_0) = s_0 \tilde{x}(s_0) + c \leq 0$, hence $Q$ has a root in the interval $[0, s_0]$ and $s_0^* \leq s_0$. On the other hand, $s_1 > s_0$, so (44) implies that $s_1$ should greater or equal to the larger root $s_1^*$. Now we lower bound $s_1^*$. Defining $G = \frac{\tilde{x}(s_0) - (2w+1)s_0}{2(2w+1)}$, completing the square for (45) gives

$$(s_1^* + G)^2 = G^2 - \frac{c}{2w + 1} \geq \left(G - \frac{2c}{(\tilde{x}(s_0) - (2w + 1)s_0)}\right)^2.
\tag{46}$$

Note that for the last inequality in (46) to hold, we need to show

$$-G \geq \frac{2c}{(-\tilde{x}(s_0) + (2w+1)s_0)},$$

or equivalently

$$(-\tilde{x}(s_0) + (2w+1)s_0)^2 \geq 4c(2w+1).$$

But from $x \leq -w$, we get $(-\tilde{x}(s_0) - (2w+1)s_0)^2 \geq (w-2c)^2$, which follows from the assumption $w \geq 18c + 2$. Now based on (46) and using $\tilde{x}(s_0) \leq -w + 2c$ and $s_0 = c/(w+1) \leq c$, we have for the larger root $s_1^*$:

$$s^* \geq s_1 \geq s_1^* \geq -2G + \frac{2c}{\tilde{x}(s_0) - (2w+1)s_0}$$

$$\geq -2G + \frac{2c}{-w + 2c - 2c + s_0}$$

$$\geq -2G - \frac{2c}{w - c}$$

$$\geq \frac{-x - 4c}{2w + 1} - \frac{2c}{w - c}.$$

But note that

$$w - c \geq \frac{w}{2} + \frac{1}{4} + \frac{w}{2} - c - \frac{1}{4} \geq \frac{2w+1}{4}.$$

Hence

$$s^* \geq \frac{-x - 12c}{2w + 1}.$$

Therefore, from (40),

$$|\tilde{x}(s^*)| \leq |x| + c - (2w + 1 - \epsilon)s^*$$

$$\leq |x| + c - (2w + 1 - \epsilon)\frac{-x - 12c}{2w + 1} \leq 13c + \epsilon \frac{-x - 12c}{2w + 1} \leq 13c + \epsilon. \quad (47)$$

where the last inequality follows from $-x \leq 2w$. Combining Equations (43) and (47), it is enough to show the following to prove $\tilde{x}(1) > 0$:

$$\frac{2w + 1 - \epsilon - |x| - c}{2} + \frac{c}{2} \geq 13c + \epsilon.$$

Using $x \geq -2w + 26c$ it suffices to show $\epsilon \leq \frac{1}{2}$. But unwrapping the definition of $\epsilon$, this is equivalent to showing

$$w\frac{2}{e^{2c} + 1} \leq \frac{1}{2}.$$

From the definition $c = \ln(w)$, this inequality holds for $w \geq 100$. $\square$

**Lemma 8** (Moving the mass to $\Theta(\sqrt{w})$)**.** *Assume $w \geq 0$. Then for $\tilde{x}(0) = x \geq -\sqrt{w+1}/2$ we have*

$$\tilde{x}(1) \geq \sqrt{w+1}/4.$$

*Proof.* Below, we will use the fact that $w \geq 0$ implies $\frac{1}{\sqrt{w+1}} \leq 1$.

As in the previous proof, note that for $\tilde{x}(s) \leq 0$ we have

$$\tilde{x}'(s) \geq w + 1.$$

Hence if $s_0$ is the first time that $\tilde{x}(s)$ reaches zero, we have $s_0 \leq \frac{1}{2\sqrt{w+1}}$. Note that for $\tilde{x}(s) \geq 0$ we have $\tilde{x}'(s) \leq w+1$, so for $s \leq s_0 + \frac{1}{2\sqrt{w+1}}$ we have $\tilde{x}(s) \leq \sqrt{w+1}/2$. But since $\tanh(1/2) < 1/2$, we get that if $s \leq s_0 + \frac{1}{2\sqrt{w+1}} \leq \frac{1}{\sqrt{w+1}}$, then $\tanh(s\tilde{x}(s)) < \frac{1}{2}$. Therefore, we get that for all $s \leq s_0 + \frac{1}{2\sqrt{w+1}}$,

$$x'(s) \geq w + 1 - \frac{w}{2} > \frac{w+1}{2}.$$

As a result, for $s_1 = s_0 + \frac{1}{2\sqrt{w+1}} \leq 1$, we see that

$$\tilde{x}(s_1) \geq \tilde{x}(s_0) + (s_1 - s_0)\frac{w+1}{2} \geq \frac{\sqrt{w+1}}{4},$$

which completes the proof, as $\tilde{x}(1) \geq \tilde{x}(s_1)$ because the particle continues to move to the right as long as it is positive. □

We can now conclude the proof of the main result of this section:

*Proof of Theorem 5.* The first inequality follows from Lemma 7 as for the standard normal we have $\mathbb{P}(x \leq -2w + \ln(w)) \leq e^{-\Theta(w^2)}$. The second inequality follows from Lemma 8, as $\mathbb{P}(x \leq -\sqrt{w+1}/4) \leq e^{-\Theta(w)}$. □

## C  Additional experiments

### C.1  Gaussian experiments

Here we consider a 2-D version of the mixture distribution of Theorem 2, i.e. $p^{(1)} = N(1,1) \otimes N(0,1)$ and $p^{(-1)} = N(-1,1) \otimes N(0,1)$. We follow the exact same experimental setup as Section 3.1 and generate 500 samples from the conditional distribution $p(x \mid z = +1)$ with varying levels of guidance.

We once again plot the mean of the probability flow ODE trajectories (projected on to the $x$-coordinate) for increasing guidance parameter values alongside the final produced samples from each trajectory in Figure 5. The figure is analogous to Figure 2, except for larger choices of the guidance parameter $w$ and the fact that the proportion of "good" samples here is the proportion of samples that did not result in NaNs (since we are no longer in the compact support setting).

We use larger $w$ values to better illustrate the behavior predicted in Theorem 2. As can be seen from Figure 5 (a), we produce more samples with larger (positive) $x$-coordinates as we increase $w$. However, we also get significantly more numerical instability, and as a result the mean trajectory plot in Figure 5 (b) is much less meaningful than it was in Figure 2.

### C.2  MNIST experiments

In Figures 6 to 14 we collect the MNIST experiments considering every other possible one-vs-all reduction. As mentioned in Section 3.2, they have near-identical behavior to the experiments of Figure 3.

### C.3  ImageNet experiments

Figures 15 to 18 show the results of repeating the experimental setup of Section 3.3 for different choices of the positive class, and also illustrate limitations of this experimental setup in the context of ImageNet.

In Figures 15 to 17, we see approximately the same behavior as in 4. Namely, guidance values for which we have support error lead to distorted samples. Similar to Figure 4, the choice $s = 5$ works well in Figure 15. However, for Figures 16 and 17, we see that we have non-zero support error even for $s = 5$. In these latter two cases, we expect the qualitatively best choice of guidance to be somewhere between $s = 1$ and $s = 5$.

Figure 18 demonstrates some limitations of our experimental setup for classes which have high levels of noise/variance. Indeed, we see that for the "basketball" class the support error is not even monotonically increasing with the guidance parameter, and that sample quality is poor across guidance levels. Furthermore, the projected trajectories become progressively more negative as opposed to positive, indicating that the direction between the means of the positive class samples and the negative class samples is likely useless in this case. Despite these various issues, there appears to still be some positive correlation between support error and sample distortion.

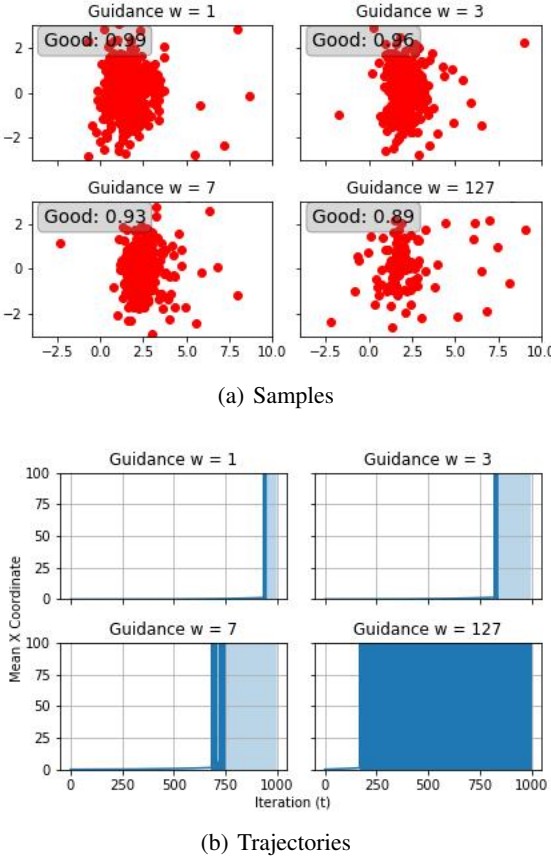

(a) Samples

(b) Trajectories

Figure 5: Mixture of Gaussians analogue to Figure 2. Proportion of good samples corresponds to non-diverged samples. Some trajectories explode due to numerical instability, leading to less meaningful mean projected trajectory plots.

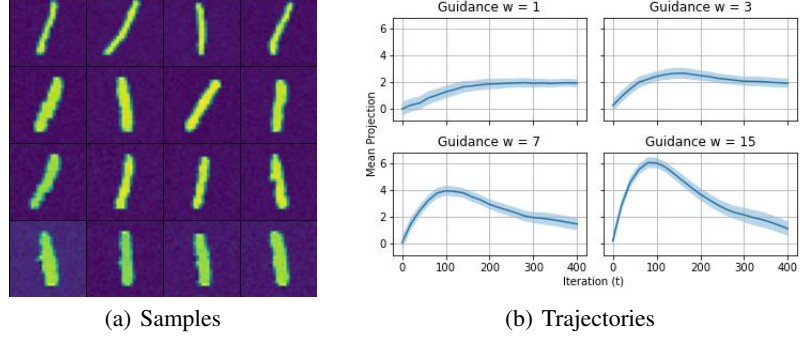

(a) Samples          (b) Trajectories

Figure 6: Experiments of Figure 3 but with the positive class fixed to be 1.

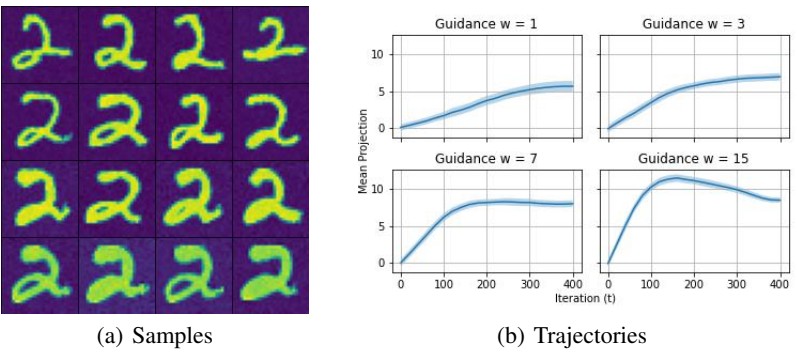

(a) Samples                    (b) Trajectories

Figure 7: Experiments of Figure 3 but with the positive class fixed to be 2.

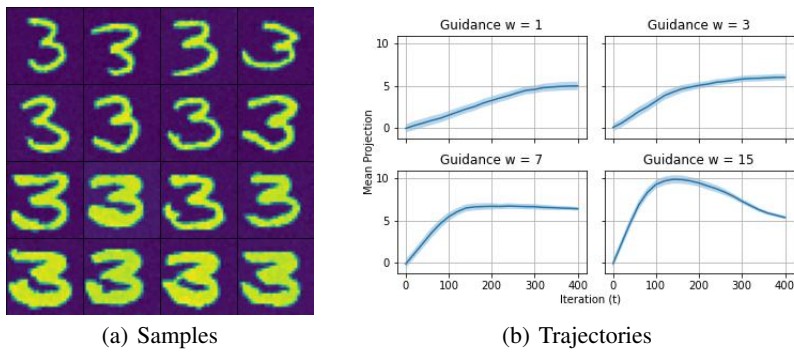

(a) Samples                    (b) Trajectories

Figure 8: Experiments of Figure 3 but with the positive class fixed to be 3.

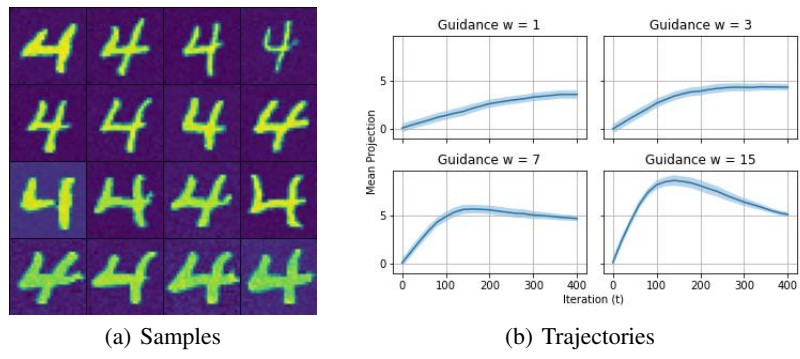

(a) Samples                    (b) Trajectories

Figure 9: Experiments of Figure 3 but with the positive class fixed to be 4.

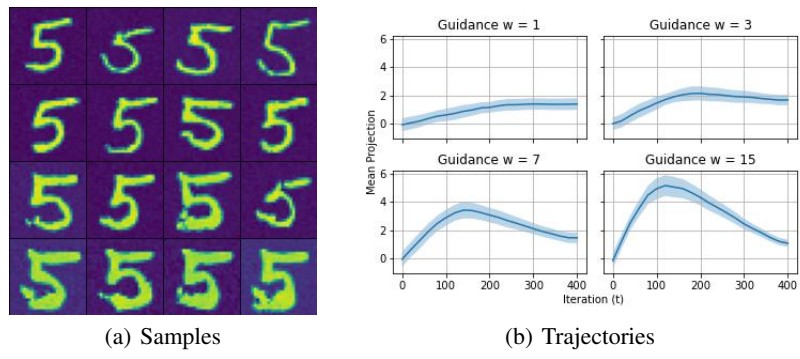

(a) Samples                    (b) Trajectories

Figure 10: Experiments of Figure 3 but with the positive class fixed to be 5.

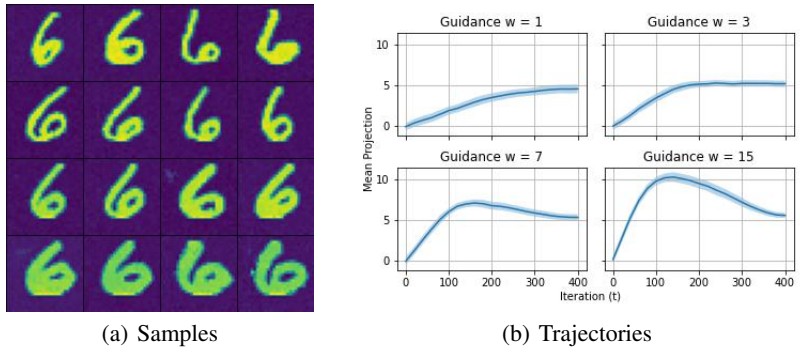

(a) Samples    (b) Trajectories

Figure 11: Experiments of Figure 3 but with the positive class fixed to be 6.

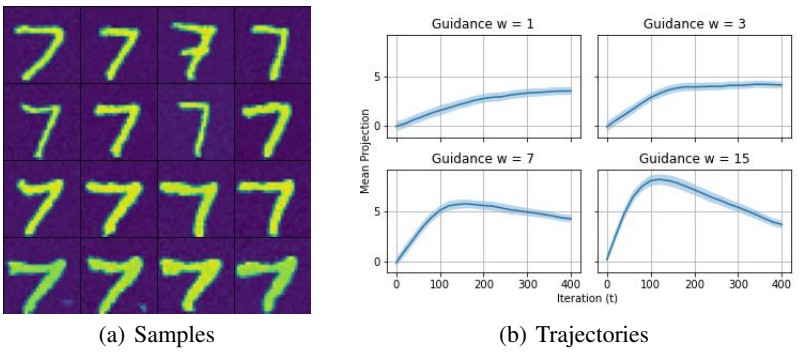

(a) Samples    (b) Trajectories

Figure 12: Experiments of Figure 3 but with the positive class fixed to be 7.

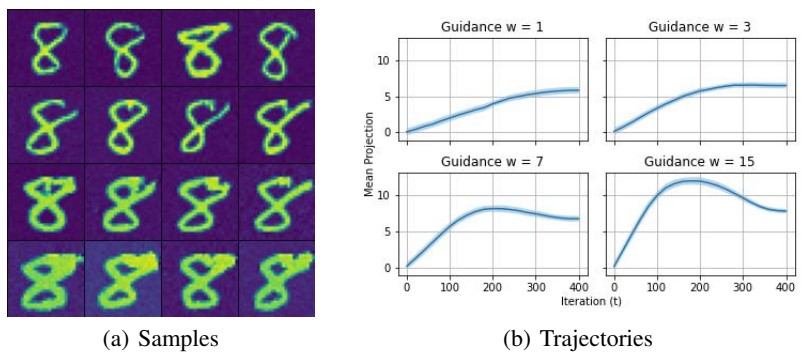

(a) Samples    (b) Trajectories

Figure 13: Experiments of Figure 3 but with the positive class fixed to be 8.

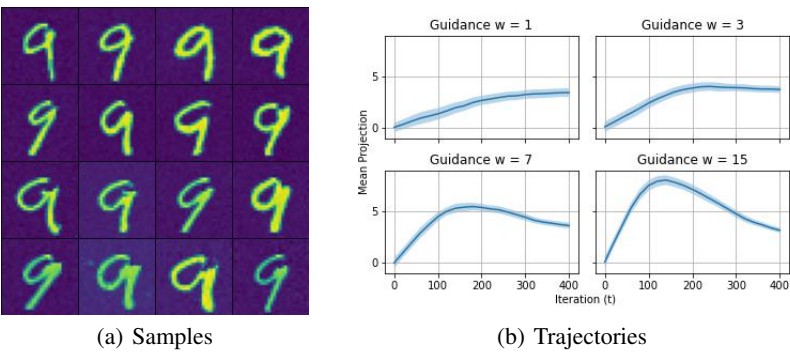

(a) Samples    (b) Trajectories

Figure 14: Experiments of Figure 3 but with the positive class fixed to be 9.

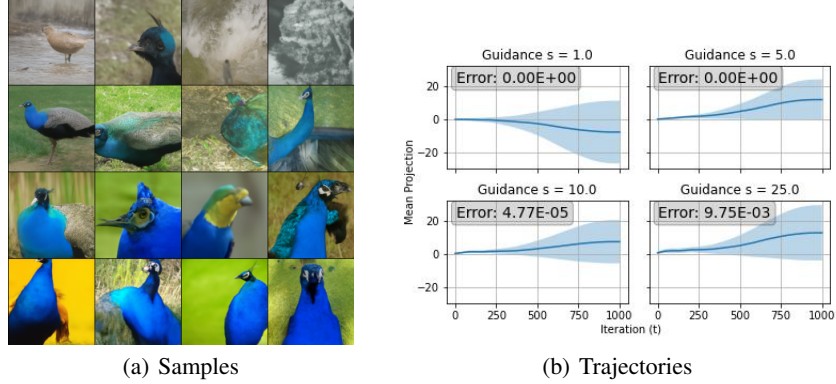

(a) Samples

(b) Trajectories

Figure 15: Experiments of Figure 4 except with the positive class taken to be 84 (peacock).

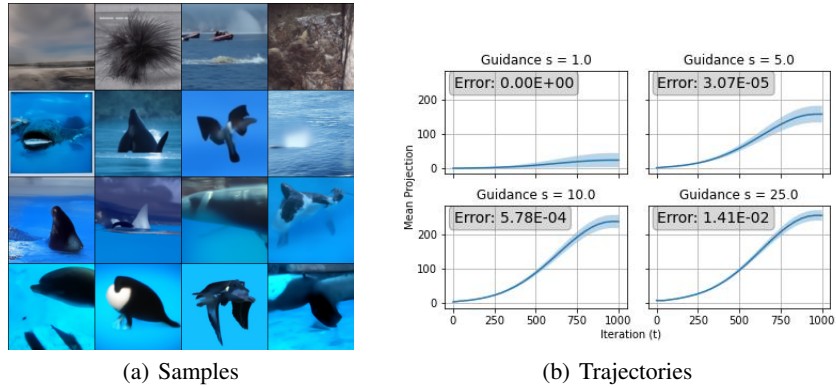

(a) Samples

(b) Trajectories

Figure 16: Experiments of Figure 4 except with the positive class taken to be 148 (killer whale).

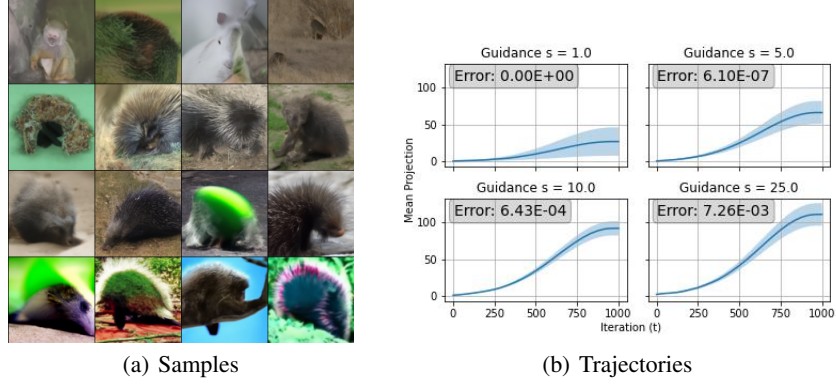

(a) Samples

(b) Trajectories

Figure 17: Experiments of Figure 4 except with the positive class taken to be 334 (porcupine).

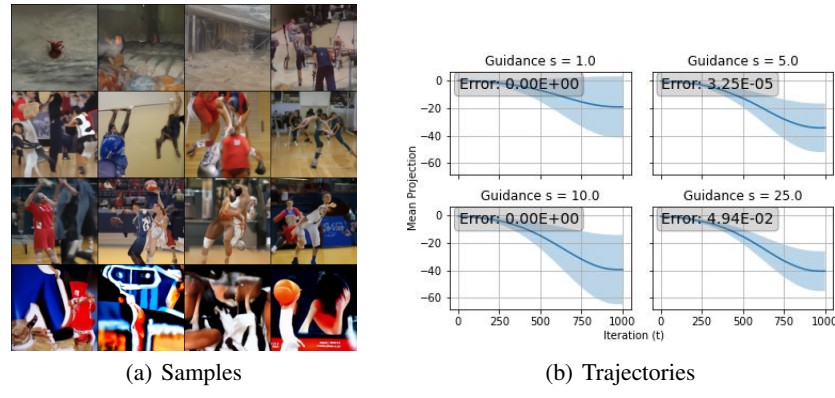

(a) Samples             (b) Trajectories

Figure 18: Experiments of Figure 4 except with the positive class taken to be 430 (basketball).

