# OpenReview forum: "What does guidance do? A fine-grained analysis in a simple setting"
_NeurIPS.cc/2024/Conference — NeurIPS 2024 poster_

### Official Review · Reviewer_4NtD · 2024-07-06

**Soundness:** 3
**Presentation:** 2
**Contribution:** 2
**Rating:** 4
**Confidence:** 3

**Summary:**

The paper characterizes the distribution from which diffusion guidance samples. It proves that guided diffusion sampling tends towards the edges of the supports of the class-conditional distributions in scenarios involving mixtures of uniform or Gaussian distributions.

**Strengths:**

* The paper clearly proposes the focused question, to characterize the distribution diffusion guidance sampling from.

* The paper clearly introduces the relationship with prior works on diffusion guidance and conditional diffusion models.

**Weaknesses:**

* The paper's structure needs improvement. The experiments should be settled at the end of the paper, rather than in Section 3. The motivation for theories could be shorter. The current placement interrupts the coherence of the theoretical discussion.

* It is not clear which main results are explained in Section 4. While it discusses the convergence speed towards $p^{(1)}$ on different cases​, it lacks solid support.

* The role of score estimation error isn't described clearly. Do Theorems 1 and 2 require ground truth scores?

* The presentation of theoretical results is not coherent. For example, there is no definition of $\tilde{x}(1)$ when it first appears in Theorem 1.

* Assumptions 1 and 2 are the same, but Theorem 4 cites Assumption 2, which is in the appendix, causing confusion.

* Typo: Line 35, reference missing.

**Questions:**

Please see weaknesses.

**Limitations:**

The consideration of only two classes for the condition and uniform and gaussian distribution is kind of simplistic.

---

> ### Author Rebuttal · Authors · 2024-08-07
>
> We would like to thank the reviewer for taking the time to review our paper as well as the helpful feedback. We hope to address the main concerns below.
>
> **Weaknesses/Questions**
>
> > The paper's structure needs improvement. The experiments should be settled at the end of the paper, rather than in Section 3. The motivation for theories could be shorter. The current placement interrupts the coherence of the theoretical discussion.
>
> We thank the reviewer for their organizational suggestions. Please see our joint response for a discussion on this, and in particular the changes we plan to make to the organization.
>
> > It is not clear which main results are explained in Section 4. While it discusses the convergence speed towards $p(1)$ on different cases​, it lacks solid support.
>
> The rigorous support for our theory is provided in the appendix. Due to space constraints and the technically intricate nature of our proofs of Theorems 1 and 2, we chose to opt for a higher-level discussion. Our reasoning was that the role of guidance is of broad interest to the diffusions community and thus merits a more conceptual approach to exposition which we will build upon in our forthcoming re-organization. We also thank the reviewer for their helpful suggestion to clarify which parts of the discussion in Section 4 correspond to which Theorems.
>
> > The role of score estimation error isn't described clearly. Do Theorems 1 and 2 require ground truth scores?
>
> Theorems 1 and 2 assume ground truth scores, and we quantify the effect of score estimation error in Theorem 3, which we originally omitted from the discussion in the main body due to space constraints. Given the cuts we plan to make to Section 5 as outlined in our joint response on organizational changes, this will free up space to include additional discussion about Theorem 3 in the main body.
>
> > The presentation of theoretical results is not coherent. For example, there is no definition of $\tilde{x}(1)_1$ when it first appears in Theorem 1.
>
> We thank the reviewer for pointing this out and will omit the notation of $\tilde{x}(1)_1$ in the theorem statements in Section 1. This notation is just meant to denote the first coordinate of the final sample, which is already clear from the rest of the text in Theorems 1 and 2 and thus unnecessary.
>
> > Assumptions 1 and 2 are the same, but Theorem 4 cites Assumption 2, which is in the appendix, causing confusion.
>
> We apologize for this oversight. This was caused by an improper compilation of the main body + supplement within the same TeX file, and will be addressed in our revision.
>
> > Typo: Line 35, reference missing.
>
> Thank you for catching this.
>
> We hope the above addresses your main concerns and we are happy to engage further to address any additional questions you may have.

---

### Official Review · Reviewer_CopY · 2024-07-13

**Soundness:** 3
**Presentation:** 3
**Contribution:** 3
**Rating:** 7
**Confidence:** 3

**Summary:**

The paper offers a theoretical investigation of the use of guidance in diffusion models. Through two stylized models, the paper fully characterizes the behavior of using guidance, which violates the commonly adopted intuition.

**Strengths:**

The paper focuses on an important question, i.e., using guidance in diffusion models, revealing an overlooked phenomenon through rigorous theoretical treatment. The illustrated phenomenon is likely to have large effects on practice, which I think is a major contribution.

**Weaknesses:**

1. The phenomenon revealed in this paper is restricted to highly stylized models, and it is unclear whether it is generally applicable.
2. The main message of this paper is the potential failure of using guidance in diffusion models. However there are no rigorous recommendations for implementing the method (e.g., the recommended choice of w is heuristic).

**Questions:**

As mentioned in the "Weaknesses" section, I wonder:
1. How general are the phenomena revealed in the stylized examples? Would it be possible to investigate it at least through simulations?
2. Would it be possible to provide a concrete implementation of the use of guidance with theoretical guarantees (even under the stylized models)?

**Limitations:**

The authors have addressed the limitations.

---

> ### Author Rebuttal · Authors · 2024-08-07
>
> We would like to thank the reviewer for taking the time to review our paper, as well as the positive encouragement and helpful feedback. We hope to address the main weaknesses/questions below.
>
> **Weaknesses/Questions**
>
> > How general are the phenomena revealed in the stylized examples? Would it be possible to investigate it at least through simulations?
>
> The phenomena we reveal are quite general. The stylized experiments in the paper are meant to conform to our theoretical setting, but similar behavior can be observed even when we relax constraints. For example, even though Theorem 1 concerns compact distributions with separated support (at least in the coordinate of interest), the empirical results of Section 3.1 hold true even when we allow the supports to overlap. Furthermore, as Theorem 1 suggests, we can consider significantly more complicated distributions than uniform over an interval; we tried experiments where we sampled from various convex bodies with and without overlap and this behavior of moving to edges of the distribution still appears. When we revise, we will include at least a subset of these experiments in the appendix.
>
> > Would it be possible to provide a concrete implementation of the use of guidance with theoretical guarantees (even under the stylized models)?
>
> Unfortunately this is a big open question even if the tilt is Gaussian, which corresponds to the heavily studied setting of posterior sampling for linear inverse problems. For the compactly supported setting we consider, note that it is trivial to sample from the tilted distribution: it is simply the distribution over the compact support selected by the classifier, and this is independent of the choice of guidance parameter.
>
> We hope the above provides more clarity, and we are happy to answer any further questions you may have.

---

> > ### Comment · Reviewer_CopY · 2024-08-12
> >
> > Thank you for the response! I remain positive about this submission.

---

### Official Review · Reviewer_gYu8 · 2024-07-13

**Soundness:** 3
**Presentation:** 2
**Contribution:** 3
**Rating:** 5
**Confidence:** 3

**Summary:**

This paper discusses the impact of diffusion guidance, especially when noting that the guided score function does not correspond to that of tilted distributions. The authors theoretically justify that a large guidance scale can lead to low-entropy and "extreme" samples. The authors further discuss score estimation in the real world and propose that a sufficient large guidance scale is more likely to lead to a sample outside the distribution domain. The authors finally introduce experiments to discuss the optimal choice of guidance scale and show that a guidance scale that is too large introduces swinging away from the support of the data distribution.

**Strengths:**

1. The paper focuses on a frontier research field, diffusion guidance, and provides systematic analysis.
2. Theories are well justified, make sense, and well explained.
3. Theoretical analysis is combined with experiments, which makes the paper more convincing.

**Weaknesses:**

1. **The introduction of the paper could be better organized.**
The authors organize the paper in a way that the introduction is a bit unclear. The current introduction consists of the background, the main results, and the related works. It would be better to separate them in individual sections. Also, the authors could provide a summary of contributions in the introduction.
2. **There seems to be uncomplete or missing parts in the paper.**
   - The conclusion section is missing.
   - The limitation and broader impact are not discussed.
   - Line 245: the authors mention the choice of positive labels but the appendix does not provide the details.

**Minors:**
   - Figure 1-4: graphics are not vectorized.
   - Line 35: missing reference.

**Questions:**

1. **How applying noise and tilting the distribution do not commute?**
   In Line 37-39, the authors mention that applying noise and tilting the distribution do not commute. Since this is a crucial point for the paper, could the authors provide more details or intuitions to explain this?

2. **How experiments on ImageNet are related to the Gaussian setting?**
   In Line 260-262, the authors discuss that the dynamics of "farther movement" resemble to those of the Gaussian setting, instead of that of MNIST. Seemingly the authors do not provide details of the experiments of Gaussian setting in the paper. Could the authors provide more details on this?

**Limitations:**

The limitation and broader impact are not discussed.

---

> ### Author Rebuttal · Authors · 2024-08-07
>
> We would like to thank the reviewer for taking the time to review our paper and the helpful feedback. We hope to address the main concerns below.
>
> **Weaknesses**
>
> > **The introduction of the paper could be better organized.** The authors organize the paper in a way that the introduction is a bit unclear. The current introduction consists of the background, the main results, and the related works. It would be better to separate them in individual sections. Also, the authors could provide a summary of contributions in the introduction.
>
> We thank the reviewer for their helpful organizational suggestions. We will separate the background (lines 14 to 46), motivating example and main results (lines 47 to 110), and related works (Lines 111 to 148) in Section 1 into three separate subsections. For the second of these three subsections where we describe main results, we will add a concluding section where we summarize our main contributions, namely 1) two toy settings which illustrate that diffusion guidance fails to sample from the tilted density in two different ways, 2) theory illustrating the impact of score estimation error on the behavior of guidance, and 3) synthetic and real data experiments validating our theory and offering practical prescriptions.
>
> > **There seems to be uncomplete or missing parts in the paper.**
>
> Thank you for pointing out the missing MNIST experiments in the appendix and we sincerely apologize for the appendix being cut off in the submission - please see the joint response that includes the majority of the cut off MNIST figures (some left out due to single page limit) which affirm line 245. We also address the issue of the conclusion section in our joint response on organizational changes we will make in the final manuscript.
>
> **Questions**
>
> > How applying noise and tilting the distribution do not commute? In Line 37-39, the authors mention that applying noise and tilting the distribution do not commute. Since this is a crucial point for the paper, could the authors provide more details or intuitions to explain this?
>
> Please see the joint response for more details.
>
> > How experiments on ImageNet are related to the Gaussian setting? In Line 260-262, the authors discuss that the dynamics of "farther movement" resemble to those of the Gaussian setting, instead of that of MNIST. Seemingly the authors do not provide details of the experiments of Gaussian setting in the paper. Could the authors provide more details on this?
>
> In the Gaussian setting, we show in Theorem 2 that as we take the guidance parameter $w$ to be large, points move towards $\pm \infty$ depending on the guided class. In this case there is no pullback effect like we show in the compactly supported setting. What we observe in lines 260-262 is that this same kind of behavior appears in the ImageNet experiments, where the sampled points move further and further along the mean-separating direction.
>
> We originally excluded the Gaussian experiments simply for space constraints in the main body; an empirical verification of Theorem 2 is also available in the joint response PDF in addition to the MNIST experiments. We apologize for excluding these originally - thank you for pointing out that it would be useful to have them.
>
> We hope the above discussion addresses your main concerns, and we are happy to answer any further questions you may have.

---

> > ### Comment · Reviewer_gYu8 · 2024-08-13
> >
> > Thank you for the author's rebuttal. Given the current state of the manuscript, I believe it requires more significant revisions than initially planned. For this reason, I cannot assign a higher score at this time and will maintain my original score.

---

### Official Review · Reviewer_xts9 · 2024-07-14

**Soundness:** 4
**Presentation:** 4
**Contribution:** 4
**Rating:** 9
**Confidence:** 4

**Summary:**

Previous authors show that tilting the score at any given noisey time t corresponds to the score of a tilted-at-that-time-t distribution, and they use this to motivate conditional sampling algorithms, but it is shown here that this is not the score of the noised version of the titlted-at-time-0 distribution (which is the one intended to sample from)

As such, there's no reason to believe priori that the guided samplers are sampling from the intended titled conditionals.
They make rigorous their observation that when you want to sample X|A but the current particle during sampling is currently close to fulfilling event B != A, the particle gets repelled at maximum velocity (under some constraints) away from set B and towards set A, and due to some weird dynamics, particles end up getting stuck on the edges of the support of X|A.

To analyze what's going on the authors consider some simple low dimensional examples as well as MNIST + Imagenet, and then provide theorems that make rigorous the above phenomena.

The authors connect the theory well to previous works on this topic (Wu et al) and in general this is an insightful read + carefully executed theory wise and experimentally.

**Strengths:**

See above summary for strengths. In short, I'd like to add that too much diffusion + generative models literature focuses too much on showing that one particular setup is able to achieve a good result on a dataset.  On the other hand, this work adds much needed questioning about what common empirical choices are doing (at best, at optimum, in any situation, etc)

**Weaknesses:**

Nothing notable.

minor:
- explain second equality in (1) to the reader (that the coef. gets normalized out + dropped due to grads)
- broken ref right before (2)

**Questions:**

Nothing notable for now, but I will add additional comments when some questions come up during the discussion period.

**Limitations:**

Nothing notable.

---

> ### Author Rebuttal · Authors · 2024-08-07
>
> We thank the reviewer for their positive feedback and for finding our work to be an insightful read with carefully executed theory and experiments. We are encouraged that they agree it is important to question what common empirical choices in the practice of diffusions are actually doing. We will also incorporate the small fixes that were suggested. We are also happy to answer any additional questions that may come up during the discussion period.

---

### Official Review · Reviewer_5E9W · 2024-07-15

**Soundness:** 3
**Presentation:** 2
**Contribution:** 3
**Rating:** 5
**Confidence:** 3

**Summary:**

This paper explores the mathematical basis for the principle of "guidance" in generative models built out of dynamical transport of measure and provides a mathematical analysis on why certain effects are empirically observed.

They provide this theoretical analysis for a mixture distribution and then test if these results hold in the case of images for classifier and classifier-free guidance.

**Strengths:**

This paper motivates well what its aims are. In addition, the authors provide a suite of experiments ranging from simple synthetic examples to support the main theoretical claims about the evolution of the probability flow ODE under different guidance scales.

**Weaknesses:**

The reviewer finds the paper pretty disorganized, to the point that it is hard to follow the validity of some of the theoretical statements as well as their implication. In particular, I'd like to draw the following comments to the authors in hopes that they can improve these aspects of the paper:

- There are some statements early on that I found confusing, and without theoretical justification. For example, the statement: "In other words, the operation of applying noise to p and the operation of tilting it in the direction of the conditional likelihood do not commute," confuses the reviewer, in the sense that it is not clear why the relation they refer to breaks down. There are proofs in other papers that show that there is a valid transport equation for the classifier guidance setting, e.g. the appendix C in [1]. The question the reviewer thinks the authors should be trying to ask is how to interpret this tilted density (their first unmarked equation).
- The organization of the theorems in the paper makes them a bit hard to follow. The authors introduce Theorems 1 and 2 early on in the motivation of the paper, but don't provide a preliminaries section until 2 pages later that try to introduce some of the distributions under consideration. Following this, there is a section on numerical experiments to motivate these theorems, but then a return to results on the mixture of uniform distributions relevant for the theorems. This organization needs serious work to be compelling. It's pretty hard to follow which aspects of the theoretical results one should be keeping track of to see if the experiments really support them. The paper then ends with a high level sketch of this last proof.
- Certain equations are introduced with no clarification of notation, for example the probability flow ODE (eq 4), nor is it clear where this equation comes from unless you know the literature. It's also unclear why a different formulation of it is included in Lemma 1.
- The reviewer appreciates the efforts of the authors to include results on image generation, however the experimentation is a bit thin and heuristic, only relying on this pullback effect. For example, on the MNIST experiments, how do you quantify this as outside the support of the density?



[1] Ma et. al. (2024) https://arxiv.org/pdf/2401.08740

**Questions:**

Can you please clarify what you mean by this notion of noising a distribution *p* (which doesn't really make sense to me, I think you mean noising the samples e.g. convolving *p* with a Gaussian) and tilting not commuting? I don't fully understand the point still. There is a valid transport equation for *p_t* and therefore also an associated probability flow, so it's a bit unclear to me still by what you mean. See the above paper.

**Limitations:**

The authors have not addressed many limitations of this work, though the reviewer does sympathize with not having easy access to compute or the image models necessary to really do some good experimentation.

---

> ### Author Rebuttal · Authors · 2024-08-07
>
> We would like to thank the reviewer for taking the time to review our paper and for the helpful feedback. We apologize for some of the organizational issues, and hope to address the main weaknesses below.
>
> **Weaknesses**
>
> > There are some statements early on that I found confusing, and without theoretical justification. For example, the statement: "In other words, the operation of applying noise to p and the operation of tilting it in the direction of the conditional likelihood do not commute," confuses the reviewer, in the sense that it is not clear why the relation they refer to breaks down.
>
> Regarding tilting and noising, please see the joint/global response where we have further clarified what we mean by noising and tilting.
>
> > There are proofs in other papers that show that there is a valid transport equation for the classifier guidance setting, e.g. the appendix C in [1]. The question the reviewer thinks the authors should be trying to ask is how to interpret this tilted density (their first unmarked equation).
>
> It appears to us that Appendix C of Ma et al. (2024) only shows that the heuristic derivation of the probability flow ODE extends to the more general flow models that they consider. In fact, the only formal claim made in Appendix C is that the guided drift used at time $t$ is the score of what we refer to in the joint response as the noised-then-tilted distribution, and what they refer to as the “tempered distribution.” In light of the argument we provide in the joint response and in our paper, however, the noised-then-tilted distribution is different from the tilted-then-noised distribution, and **guidance provably does not sample from the intended tilted distribution**. In our original submission, we also gave a very simple example in Lines 47-54 supporting this claim.
>
> > The organization of the theorems in the paper makes them a bit hard to follow. The authors introduce Theorems 1 and 2 early on in the motivation of the paper, but don't provide a preliminaries section until 2 pages later that try to introduce some of the distributions under consideration. Following this, there is a section on numerical experiments to motivate these theorems, but then a return to results on the mixture of uniform distributions relevant for the theorems. This organization needs serious work to be compelling. It's pretty hard to follow which aspects of the theoretical results one should be keeping track of to see if the experiments really support them. The paper then ends with a high level sketch of this last proof.
>
> We thank the reviewer for their organizational suggestions. Please see our joint response for a discussion on this, and in particular the changes we plan to make to the organization.
>
> > Certain equations are introduced with no clarification of notation, for example the probability flow ODE (eq 4), nor is it clear where this equation comes from unless you know the literature. It's also unclear why a different formulation of it is included in Lemma 1.
>
> Lemma 1 was useful in the proof in the supplement as it gives a more explicit form for the drift of the ODE in the two-component Gaussian mixture setting that we consider. However, in our reorganization (see joint response), we will omit it in favor of a more clear exposition of the probability flow ODE in the introduction and defer the details of Lemma 1 to the supplement.
>
> > The reviewer appreciates the efforts of the authors to include results on image generation, however the experimentation is a bit thin and heuristic, only relying on this pullback effect. For example, on the MNIST experiments, how do you quantify this as outside the support of the density?
>
> For the MNIST experiments, we use the notion of class support more loosely than in the case of the synthetic experiments, since we do not have a precise definition to work with. We more so meant to draw attention to the fact that increasing the guidance parameter leads to sampling trajectories that move further and further along the separating direction between a fixed class and all other classes (visualized in Figure 3), and that this correlates with qualitatively worse samples. When we revise, we will add plots of the intermediate samples corresponding to when the trajectory has moved very far out to provide better intuition for the undesirable effects of large w in these cases.
>
> **Questions**
>
> The main question is addressed above in the first two points under weaknesses.
>
> We hope the above addresses the main concerns of the review, and we are happy to engage further and answer any additional questions you may have.

---

> > ### Comment · Reviewer_5E9W · 2024-08-08
> > **Response to rebuttal**
> >
> > Thanks kindly to the authors for the substantive response. It took a day for the general reply to load but I see it now.
> >
> > The clarification regarding the mechanism of noising the distribution, as well as its disparity from the tilted distribution is now nice and clear to me. I hope that the authors agree that being explicit like this in the paper will make it much more legible and insightful. I am happy to raise my score given the proposed reworkings -- I really think it will help the paper!
> >
> > One little caveat -- the MNIST experiments are still pretty heuristic, but I understand the challenge in making them less so. It is something worth a bit more thought, perhaps down the road, because the theoretical analysis in this work is insightful. Thanks.

---

> > > ### Comment · Reviewer_5E9W · 2024-08-08
> > > **Updated score**
> > >
> > > One final point - I have preliminarily updated my score from a 3 to a 5. I am happy to push it farther once the changes are made!

---

> > > > ### Author Response · Authors · 2024-08-10
> > > >
> > > > We are glad that the proposed updates and clarifications have made our key message clearer, and we are grateful for your prompt response. We definitely agree that the MNIST experiments are quite heuristic - unfortunately even in simple theoretical settings it is difficult to pin down a precise recommendation for $w$ as the results tend to be asymptotic. Our main hope is that the ideas we present will at least provide some intuition for practitioners for how to think about tuning the guidance parameter without having to run a full evaluation of generated samples. And indeed, for future work we hope to think about ways to make more precise recommendations.
> > > >
> > > > We will be happy to make the proposed changes as soon as it is possible to revise the paper. Unfortunately, this is not possible during the rebuttal/discussion period to our knowledge. In the meantime, if any further questions arise we are happy to answer them. Thanks again for engaging with our work.

---

### Author Rebuttal · Authors · 2024-08-07

We would like to thank all of the reviewers for taking the time to review our paper and for all of the helpful feedback. We hope to address some common points of discussion below.

**Tilting and noising do not commute.**

We use the term “noising” a distribution $p$ to mean the distribution obtained by taking a sample $X\sim p$ from the original distribution and then putting it through the noise process, in our case, $X\_t = a\_t X + \xi\_t$, where $\xi\_t\sim N(0,b\_t I)$ is a Gaussian. We will clarify this.

To formalize the fact that tilting and convolving do not commute, we define the tilted distribution (with parameter $w$) by

$$p^{z,w}(x) \propto p(x) p(z|x)^{1+w}$$

where $(z, x)$ is drawn from a joint distribution and z is the label. The tilted-then-noised distribution is

$$p^{z,w}\_t = ((a_t)\_*p^{z,w}) * \gamma\_{b\_t}\quad (1)$$

where $\gamma\_t$ is the density of the Gaussian $N(0,b\_t I)$ and we use $a\_*p$ to denote the distribution of $aX$ where $X\sim p$. I.e., it is the distribution of $X\_t = a\_t X + \xi\_t$ where $X\sim p^{z,w}$ and $\xi\_t \sim N(0,b_t I)$. The probability flow ODE that would result in the distribution $p^{z,w}(x)$ would use the score function of $p^{z,w}\_t$, i.e., $\nabla \log p^{z,w}\_t$.

However, this is not the score function actually used in diffusion guidance, which uses the noised-then-tilted distribution

$$(p\_t)^{z,w}(x\_t) = p\_t(x\_t) p\_t(z | x\_t)^{1+w}.\quad (2)$$

Here, $p\_t = ((a\_t)\_*p) * \gamma\_t$, and the conditional density $p\_t(z | x\_t)$ is interpreted as the conditional distribution of $z$ given $x\_t$, where $x\_t$ is produced by taking $x|z$ and then letting $x\_t =a\_t x + \xi$ where $\xi\sim N(0,b\_tI)$. In words, it is the classifier for the noisy sample.

(1) and (2) are not the same in general for $w\ne 0$, because if they were, then diffusion guidance (which uses (2)), would produce the tilted distribution $p^{z,w}$; however, our analysis for compactly supported distributions shows that diffusion guidance results in samples that are not even in the support of $p^{z,w}$.

**Re-organization of results in the paper.**

We thank the reviewers for their feedback on improving the organization of the paper and believe the following changes will improve the flow and render our key message more transparent:

1) We will move the experiments section and figures to the end of the main body in order to not break the flow of the theoretical discussion.

2) We will define the probability flow ODE immediately after Line 32 instead of in the Preliminaries so that the algorithm we consider is clear to the reader early on.

3) Theorems 1 and 2 are meant to be self-contained and readable independently of the Preliminaries, hence the ordering in the submission. Indeed, in the theoretical computer science literature, it is standard to have informal theorem statements in the Introduction prior to the section introducing technical preliminaries. Nevertheless, to make Theorems 1 and 2 even more parseable, we will remove the $\tilde{x}(1)_1$ notation, and we will make it clearer that Assumption 2 is in reference to something later in the document.

4) The proof of our main result is quite technically intricate, yet we believe our contribution to ultimately be largely of a conceptual nature that is of broad interest to the practical and theoretical communities working on diffusions. With this in mind, we will significantly rework Section 5 as follows. First, we will integrate the main assumption and theorem in Section 5 into Section 4 to add supporting technical detail to the high-level discussion already present in Section 4. We will then remove the material in Lines 303 to 325 in order to make room for a conclusion section (see below) and for a lengthier discussion in the Introduction. In particular, based on certain confusions among some of the reviews regarding our main conceptual point about the fact that tilting and noising do not commute, we will allocate an extra two paragraphs in the Introduction to make this point even clearer conceptually.

5) We will include a conclusion section to provide a more thorough discussion of limitations that raise the possibility of future exploration. For instance, we will mention that while we show interesting behavior in simple toy models, it remains an important challenge to characterize the behavior of guidance in higher-dimensional, non-separable settings.

**Experiments.**

Some experiments were cut-off from the Appendix and we sincerely apologize for that, and thank the reviewers for catching that. They are included in the attached PDF, along with experiments in the synthetic Gaussian setting to complement the mixture of uniforms setting.

---

### Decision · Program_Chairs · 2024-09-25

**Decision:**

Accept (poster)

**Comment:**

This paper studies how guidance influences the generated mixture distributions. From my reading and reviewers' assessments, the theoretical results are new and nontrivial compared to the related earlier results by Wu et al.

There are one negative review and four positive review of the paper. The biggest criticism of the current version is the organization (Reviewers 5E9W, gYu8, 4NtD). The authors proposes a reorganization of the paper in the general rebuttal and it was appreciated by one of the reviewer. I believe such revision is necessary and can be a bit heavy for very short period. However, I am willing to trust the authors for making these changes possible in the final version (I would also encourage the authors to carefully proofread the manuscript at least to clear broken links, typos and grammatical issues). Other weaknesses of the paper are a lack of implications to practitioners and the theory being stated for simplified models. These charges remain valid, yet do not outweigh the contributions of the paper.

Therefore, I am recommending acceptance of the paper.